# Inflammatory macrophages reprogram to immunosuppression by reducing mitochondrial translation

Marlies Cortés [1] ✉, Agnese Brischetto [1,12], M. C. Martinez-Campanario [1,12], Chiara Ninfali [1], Verónica Domínguez[2], Sara Fernández[3], Raquel Celis[4], Anna Esteve-Codina[5], Juan J. Lozano[6], Julia Sidorova[6], Gloria Garrabou[3], Anna-Maria Siegert [7], Carlos Enrich[8], Belén Pintado[2], Manuel Morales-Ruiz [6,8,9], Pedro Castro [3], Juan D. Cañete[4] & Antonio Postigo [1,6,10,11] ✉

Acute inflammation can either resolve through immunosuppression or persist, leading to chronic inflammation. These transitions are driven by distinct molecular and metabolic reprogramming of immune cells. The anti-diabetic drug Metformin inhibits acute and chronic inflammation through mechanisms still not fully understood. Here, we report that the anti-inflammatory and reactive-oxygen-species-inhibiting effects of Metformin depend on the expression of the plasticity factor ZEB1 in macrophages. Using mice lacking *Zeb1* in their myeloid cells and human patient samples, we show that ZEB1 plays a dual role, being essential in both initiating and resolving inflammation by inducing macrophages to transition into an immunosuppressed state. ZEB1 mediates these diverging effects in inflammation and immunosuppression by modulating mitochondrial content through activation of autophagy and inhibition of mitochondrial protein translation. During the transition from inflammation to immunosuppression, Metformin mimics the metabolic reprogramming of myeloid cells induced by ZEB1. Mechanistically, in immunosuppression, ZEB1 inhibits amino acid uptake, leading to downregulation of mTORC1 signalling and a decrease in mitochondrial translation in macrophages. These results identify ZEB1 as a driver of myeloid cell metabolic plasticity, suggesting that targeting its expression and function could serve as a strategy to modulate dysregulated inflammation and immunosuppression.

Inflammation is a natural protective response to infection and tissue injury. It involves an acute inflammatory phase, which can be followed by an immunosuppressed (tolerogenic) state, wherein immune cells are unable to respond to a secondary challenge[1–3]. Immunosuppression protects tissues from excessive inflammation, but if prolonged over time, it can increase susceptibility to secondary infections. On the other hand, an incomplete or dysregulated resolution of acute inflammation and continuous proinflammatory stimuli can override immune tolerance and lead to autoimmunity and chronic inflammation[4].

The transition of monocytes/macrophages from an inflammatory phenotype toward an immunosuppressed state involves their

metabolic reprogramming[5–7]. During inflammation, there is a switch from mitochondrial oxidative phosphorylation (OxPhos) to glycolysis and lactate production. Mitochondria shift their role from ATP generation to succinate oxidation, which in turn stimulates the production of reactive oxygen species (ROS) and inflammatory cytokines[8,9]. Lactate is not only a byproduct of glycolysis but it also promotes the epigenetic activation of anti-inflammatory genes, leading to their subsequent upregulation during immunosuppression[10]. Inhibition of mitochondrial translation by antibiotics of the tetracycline family and analogues of the anti-diabetic drug Metformin reduces both bacterial lipopolysaccharide (LPS)-induced production of inflammatory cytokines in vitro and tissue damage in vivo[11,12]. Alterations in the metabolism of immune cells also play a pathogenic role in the development of autoimmune chronic inflammatory diseases (e.g., psoriasis, arthritis, colitis)[13–15].

The mechanisms by which Metformin inhibits acute and chronic inflammation are cell- and context-dependent and are still being uncovered. Metformin exerts its anti-inflammatory effects through various pathways, including inhibiting the mitochondrial electron transport chain complex I (ETC-CI) and mTORC1 signaling, as well as reducing mitochondrial ROS production[16,17]. However, in some cancer cells and CD8[+] tumor-infiltrating T cells, Metformin has been reported to stimulate ROS production[18,19]. Metformin promotes mitophagy to eliminate damaged mitochondria, thus dampening inflammasome activation[20]. Furthermore, Metformin can inhibit inflammasome activation independently of AMPK- and NFkB signaling[21,22].

The transcription factor ZEB1 enables and maintains cell plasticity in cancer cells and is best known for inducing an epithelial-to-mesenchymal transition (EMT) during embryonic development and cancer progression (reviewed in[23–26]). ZEB1 expression enhances the pro-tumoral effects of tumor-associated macrophages[27], and induces a stem-like phenotype in macrophages upon viral infection[28]. This evidence prompted us to investigate the potential role of ZEB1 in the regulation of macrophage transcriptomic and metabolic plasticity during acute and chronic inflammation.

Here, we find that ZEB1 expression is required for the inflammatory and immunosuppressive phenotypes of macrophages, playing opposing roles in both stages. In addition, we show that Metformin's anti-inflammatory and ROS-inhibiting effects in models of sepsis and psoriatic disease are dependent on the expression of ZEB1 in macrophages. Using mice lacking Zeb1 in their myeloid cells, as well as samples from human patients and mouse models of sepsis and psoriasis, we show that the diverging effects of ZEB1 in inflammation and immunosuppression are mediated through its inhibition of amino acid transport, mitochondrial protein translation and content, and its induction of autophagy. Our results suggest that Metformin pre-treatment induces an immunosuppressive-like state in inflammatory macrophages and that ZEB1 is required for the inhibition of mito-chondrial translation in immunosuppressed macrophages. ZEB1 limits acute and chronic inflammation by reducing amino acid levels and consumption in macrophages thereby inhibiting mTORC1 signaling and mitochondrial translation.

Altogether, these results identify a mechanism that regulates macrophage metabolic plasticity, presenting a potential target for modulating dysregulated inflammation and immunosuppression in sepsis and autoimmune diseases.

## Results

### ZEB1 has a dual role being required for both the induction and resolution of inflammation

To study the immunogenic-to-immunosuppressive reprogramming of macrophages in the context of acute inflammation, we used a model of acute inflammation triggered by lipopolysaccharide (LPS)[29] (see Supplementary Fig. S1A and Supplementary Methods), where mice or macrophages were divided into different groups. One group was subjected to the acute inflammation protocol (referred to as "LPS"), wherein they received the vehicle (PBS) followed by a single dose of LPS after 24 h. The other group underwent the immunosuppression protocol (referred to as "LPS + LPS"), wherein they received an initial dose of LPS, followed by a second dose of LPS 24 h later. These LPS-induced acute inflammation and immunosuppressive responses are primarily mediated by macrophages[30]. Accordingly, expression of the inflammatory cytokine IL6 in mouse peritoneal macrophages and human monocyte-derived macrophages increased during acute inflammation but to a lesser extent upon the second antigenic challenge in the immunosuppressive protocol (Supplementary Fig. S1B and S1C). We also compared IL6 expression in human peripheral blood mononuclear cells (PBMC) in different conditions. Firstly, we compared PBMCs from septic patients at ICU admission (0 h, representing an acute inflammatory state) with the PBMCs of the same patients collected three days later (72 h, representing an immunosuppressed state) (Supplementary Fig. S1D). Additionally, we compared PBMCs from healthy donors with those from patients with a chronic inflammatory disease, namely psoriatic arthritis (PsA), which affects approximately 30% of patients with psoriatic disease (Supplementary Fig. S1E). In all cases, PBMCs were either left untreated (PBS) or subjected to in vitro incubation with LPS for 2 hours. In line with their immunosuppressed state, the response of PBMCs from septic patients at 72 h to a new inflammatory challenge was only around 26% relative to the response of PBMCs from septic patients at 0 h (Supplementary Fig. S1D). In contrast, in the PBMCs of patients with PsA, IL6 increased around 500 times relative to PBMCs from healthy donors (Supplementary Fig. S1E).

The transcription factor ZEB1 is best known for promoting cellular plasticity in epithelial cells during cancer initiation and progression[24,25,31,32]. We found that ZEB1 levels increased when both human and mouse macrophages transitioned to an immunosup-pressed state (Supplementary Fig. S1F–S1H). To examine whether ZEB1 modulates the phenotype and function of macrophages during acute inflammation or immunosuppression, we generated a Zeb1[fl/fl] mouse (hereinafter referred to as Zeb1[WT] mouse) that was then crossed with Lysm[Cre] mice to delete Zeb1 in myeloid cells (hereafter referred to as Zeb1[ΔM]) (Supplementary Fig. S1I–S1K).

Zeb1[WT] and Zeb1[ΔM] mice were each divided into two cohorts and subjected to the LPS-induced lethal endotoxemia and LPS + LPS-induced immunosuppression protocols (see Supplementary Fig. S1A and Supplementary Methods). In the LPS cohort (systemic acute inflammation protocol), Zeb1[ΔM] mice exhibited greater survival than Zeb1[WT] mice (Fig. 1a). Interestingly, in the LPS + LPS cohort (immuno-suppressive protocol), the reverse pattern was found; Zeb1[ΔM] mice exhibited lower survival than Zeb1[WT] mice. The composition of myeloid cells entering the peritoneal cavity varies during the LPS response with an increase in the proportion of monocytes during the course of sepsis[33]. However, we observe no difference in the distribution of myeloid subpopulations between Zeb1[WT] and Zeb1[ΔM] mice (Supplementary Fig. S1L).

The above data suggest that ZEB1 plays opposing roles in the macrophage-mediated inflammatory and immunosuppressive responses to LPS. To define the mechanisms by which ZEB1 does so, we conducted a bulk RNA sequencing (RNAseq) of peritoneal macrophages isolated from Zeb1[WT] and Zeb1[ΔM] mice subjected to the LPS and LPS + LPS protocols (Supplementary Fig. S1M, N). In the LPS condition (acute inflammation), Zeb1[ΔM] macrophages expressed lower levels of inflammatory genes (e.g., Il1a, Il6, Nfkb1) than their Zeb1[WT] counterparts (Fig. 1b–f and Supplementary Fig. S1O). In the LPS + LPS condition (immunosuppression), inflammatory genes were expressed similarly in macrophages of both genotypes. However, a reverse pattern was observed with regard to several anti-inflammatory and homeostatic genes (e.g., Il4, Retnlg) (Fig. 1d, e, g). Although Zeb1[WT] and Zeb1[ΔM]

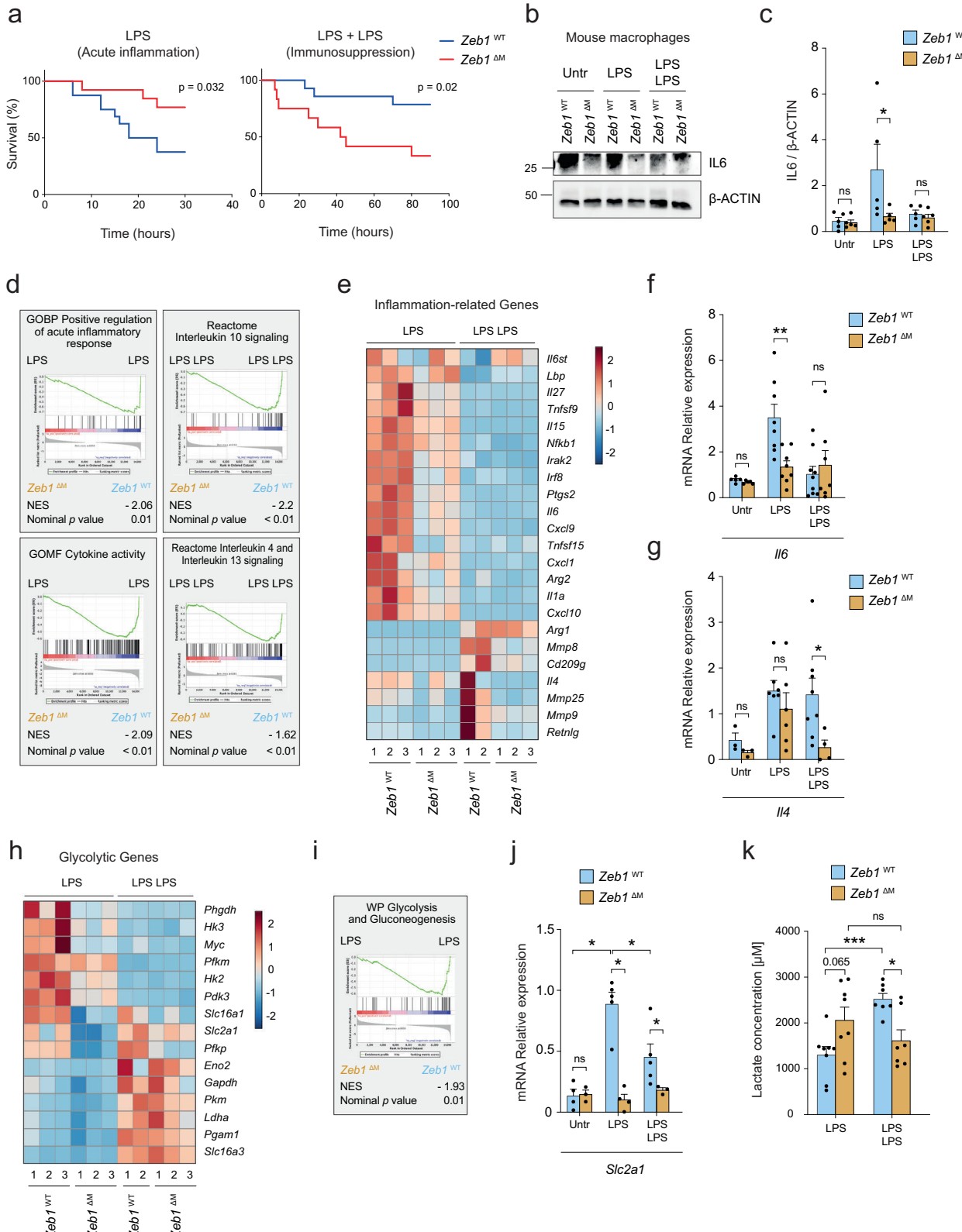

macrophages expressed similar levels of these anti-inflammatory and homeostatic genes during LPS, *Zeb1*[ΔM] macrophages expressed lower levels of those genes than *Zeb1*[WT] macrophages during LPS + LPS (Fig. 1e and g). Taken together, these findings suggest that ZEB1 plays dual roles in both the induction and resolution phases of inflammation. ZEB1 appears to modulate macrophage plasticity by upregulating

inflammatory genes during acute inflammation and homeostatic genes during immunosuppression.

## ZEB1 increases glycolysis during acute inflammation

The transition of macrophages from an inflammatory state to an anti-inflammatory state is accompanied by metabolic reprogramming,

**Fig. 1 | ZEB1 plays a dual role being required for both the induction and resolution of inflammation. a** Survival plots of 8–10 weeks old female $Zeb1^{WT}$ and $Zeb1^{\Delta M}$ mice treated with LPS (15 $Zeb1^{WT}$, 12 $Zeb1^{\Delta M}$) to induce acute inflammation, and treated with LPS + LPS (14 $Zeb1^{WT}$, 12 $Zeb1^{\Delta M}$) to induce immunosuppression. **b** Peritoneal macrophages from $Zeb1^{WT}$ and $Zeb1^{\Delta M}$ mice were untreated, or treated with LPS or LPS + LPS and blotted for IL6 along with β-ACTIN as a loading control. The blot shown is representative of five independent experiments. **c** quantification of IL6 levels relative to β-ACTIN levels in $n = 5$ biologically independent experiments as in (**b**). **d** *Left Panel:* GSEA plots for inflammatory signatures comparing macrophages from $Zeb1^{WT}$ and $Zeb1^{\Delta M}$ mice treated with LPS. *Right panel:* GSEA plots for anti-inflammatory signatures comparing macrophages from $Zeb1^{WT}$ and $Zeb1^{\Delta M}$ mice treated with LPS + LPS. **e** Heatmap of inflammation-related genes in peritoneal macrophages from $Zeb1^{WT}$ and $Zeb1^{\Delta M}$ mice treated with either LPS or LPS + LPS. **f** *Il6* mRNA levels in peritoneal macrophages from $Zeb1^{WT}$ and $Zeb1^{\Delta M}$ mice either untreated, treated with LPS, or treated with LPS + LPS ($n = 5,5,8,8,9,7$). **g** as in (f), but for *Il4* ($n = 3,3,7,6,8,4$). **h** Heatmap of glycolytic genes in peritoneal macrophages from $Zeb1^{WT}$ and $Zeb1^{\Delta M}$ mice subjected to either LPS or LPS + LPS. **i** GSEA plots for Glycolysis and Gluconeogenesis signature comparing macrophages from $Zeb1^{WT}$ and $Zeb1^{\Delta M}$ mice treated with LPS. **j** As in (f), but for *Slc2a1* ($n = 4,3,5,4,5,3$). **k** Lactate levels in macrophages from $Zeb1^{WT}$ and $Zeb1^{\Delta M}$ mice treated with LPS ($n = 8$) or LPS + LPS ($n = 7$). Statistical analysis of Kaplan Meier survival plots was assessed by the Log-rank (Mantel-Cox) test. Graph bars represent mean values +/− SEM with two-tailed unpaired Mann-Whitney test. $p \le 0.001$ (***), $p \le 0.01$ (**) or $p \le 0.05$ (*) levels, or non-significant (ns) for values of $p > 0.05$. Numerical values had been added for $0.05 < p < 0.075$. Raw data along $p$ values for statistical analyses are included in the Source Data file.

shifting from glycolysis to oxidative phosphorylation (OxPhos)[6]. Consistent with the inflammatory signature linked to ZEB1 in acute inflammation, $Zeb1^{WT}$ macrophages expressed higher levels of genes associated with glycolysis, (e.g., *Slc2a1, Hk2, Hk3, Pdk3*) compared to $Zeb1^{\Delta M}$ counterparts during acute inflammation (Fig. 1h–j). However, in the LPS + LPS condition, we found that enzymes associated with lower glycolysis that catalyze the conversion of glyceraldehyde-3-phosphate to pyruvate (e.g., *Gapdh, Pkm, Pgam1, Ldha*) were upregulated in both genotypes (Fig. 1h). In the same line, our analysis of a published array of the transcriptome of septic patients during the first week at the ICU (GSE131411)[34] indicated that ZEB1 expression correlates with inflammatory (*IL1B*) and glycolytic (*SLC2A1*) genes at the beginning of the septic process (16 h and 48 h) and with anti-inflammatory (*IL4*) and anti-oxidant (*GSS*) genes at the immunosuppressive state (Supplementary Fig. S1P).

Lactate production by macrophages in the aftermath of acute inflammation is required for the subsequent activation of anti-inflammatory genes in immunosuppressed macrophages[10,35]. In that line, we found that $Zeb1^{WT}$ macrophages subjected to LPS + LPS produced more lactate than those under LPS (Supplementary Fig. S1Q). $Zeb1^{\Delta M}$ macrophages produced less lactate than $Zeb1^{WT}$ macrophages during immunosuppression (Fig. 1k), which may contribute to the impaired anti-inflammatory transition in $Zeb1^{\Delta M}$ macrophages and the decreased survival of $Zeb1^{\Delta M}$ mice following two doses of LPS.

## ZEB1 reduces mitochondrial content during immunosuppression

It has been reported that, during acute inflammation, the expression of mitochondrial DNA (mtDNA)-encoded genes in leukocytes correlates with the severity of sepsis[36]. Interestingly, our RNAseq analysis revealed that LPS-treated $Zeb1^{WT}$ macrophages exhibited higher expression of mtDNA-encoded genes compared to $Zeb1^{\Delta M}$ macrophages (Supplementary Fig. S2A). The higher induction of IL6 in $Zeb1^{WT}$ macrophages compared to $Zeb1^{\Delta M}$ macrophages was accompanied by an upregulation of TOMM20, a nuclear DNA (nDNA)-encoded mitochondrial protein that we used as a proxy for mitochondrial content (Supplementary Fig. S2B). This suggests that the reduced inflammatory response of $Zeb1^{\Delta M}$ macrophages may be related to their altered mitochondrial function.

Peritoneal macrophages from mice either untreated (PBS) or following treatment with one or two doses of LPS were assessed for their mitochondrial content by their staining for MitoTracker™ Green (MTG). Immunosuppressed macrophages from mice treated with LPS + LPS showed lower mitochondrial content than macrophages from mice injected with PBS or a single dose of LPS (Fig. 2a). A comparable decrease in mitochondrial content—assessed by both MTG staining and MT-CO1 (mitochondrially-encoded Cytochrome C Oxidase I) expression—was also found in the immunosuppressed PBMCs of septic patients at 72 h relative to the immune-responsive PBMCs

from the same septic patients at 0 h or healthy donors (Fig. 2b–d, and Supplementary Fig. S2C).

These results prompted us to investigate whether alterations in the mitochondria content can contribute to in vivo immunosuppression in mice. We conducted a transmission electron microscopy (TEM) analysis to examine the ultrastructure of macrophages isolated from $Zeb1^{WT}$ and $Zeb1^{\Delta M}$ mice that had either been left untreated or treated with a single dose of LPS for different durations (30 min, 3 h, and 12 h), as well as with LPS + LPS. Interestingly, macrophages isolated from $Zeb1^{\Delta M}$ mice treated with LPS at 3 h contained fewer mitochondria compared to macrophages from $Zeb1^{WT}$ mice. However, when mice were treated with LPS + LPS, macrophages from $Zeb1^{\Delta M}$ mice had more mitochondria than macrophages from $Zeb1^{WT}$ mice (Fig. 2e, f). These results support the hypothesis that ZEB1 regulates mitochondria content in opposite directions in inflammation and immunosuppression.

## ZEB1 activates p62 and promotes autophagy during inflammation

The accumulation of damaged mitochondria during inflammation increases ROS production and activates inflammasome signaling, highlighting the importance of mitophagy as an important anti-inflammatory self-limiting mechanism[37,38]. Compared to $Zeb1^{WT}$ counterparts, and particularly in the LPS condition, $Zeb1^{\Delta M}$ macrophages expressed lower levels of autophagy/mitophagy-related genes (e.g., *Sqstm1, Tbc1d17, Rab9, Cisd2*) and higher levels of anti-autophagy/mitophagy ones (e.g. *Usp30*) (Fig. 2g). p62 (encoded by *Sqstm1*) binds damaged mitochondria—as well as other damaged organelles and ubiquitinated proteins—and recruits them to autophagosomes, which subsequently fuse with autolysosomes for degradation (mitophagy) in a mTORC1-dependent manner[39]. Mitophagy prevents the release of inflammasome-activating signals and limits excessive ROS production during acute inflammation[40–42].

Treatment of $Zeb1^{WT}$ and $Zeb1^{\Delta M}$ mice with LPS upregulated p62 mRNA and protein expression in $Zeb1^{WT}$ macrophages but not in $Zeb1^{\Delta M}$ counterparts (Fig. 2h, i, and Supplementary Fig. S2D). Analysis of the *SQSTM1* promoter identified several consensus sequences for ZEB1 whose capacity to recruit ZEB1 were tested in chromatin immunoprecipitation (ChIP) assays. ZEB1 bound to *SQSTM1* promoter and to a larger extent in human monocyte-derived macrophages treated with LPS than in those treated with LPS + LPS (Supplementary Fig. S2E). In addition, compared to LPS-treated $Zeb1^{\Delta M}$ peritoneal macrophages, LPS-treated $Zeb1^{WT}$ macrophages showed increased co-localization of lysosome staining (Lyso Dye™) with the Mtophagy Dye™, indicating enhanced lysosomal-mediated degradation (Supplementary Fig. S2F). Next, we examined macrophages from both genotypes treated with a single dose of LPS for signs of autophagy using TEM. $Zeb1^{WT}$ macrophages, in comparison to $Zeb1^{\Delta M}$ macrophages, exhibited a higher number of autolysosomes (Fig. 2J, yellow asterisks) containing cytosolic material, including mitochondria, indicative of damaged

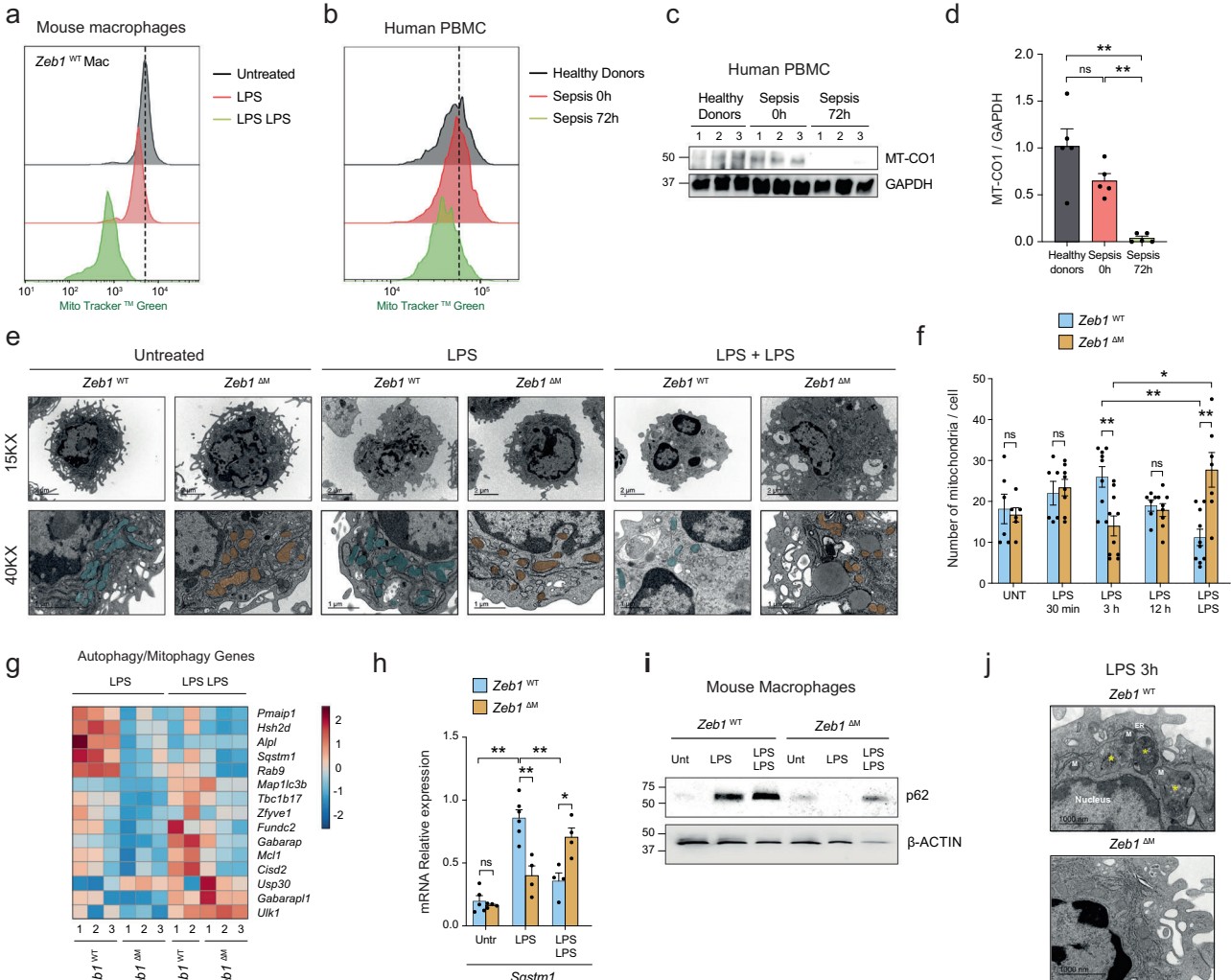

**Fig. 2 | ZEB1 regulates mitochondrial content and autophagy during inflammation. a** MTG staining in peritoneal macrophages from wild-type mice untreated, treated with LPS, or treated with LPS + LPS. The FACS plot is representative of at least 3 independent experiments. **b** MTG staining in the monocyte-enriched PBMC of healthy controls and septic patients at 0 h and 72 h. The FACS plot is representative of at least 5 independent experiments. **c** Western blot for MT-CO1 and GAPDH in PBMCs from three healthy donors and three septic patients at 0 and 72 h. The blot is representative of three independent experiments. **d** MT-CO1 protein levels in five healthy controls and five patients relative to GAPDH. **e** TEM of mitochondria in sorted macrophages from *Zeb1*^WT (labeled in blue) and *Zeb1*^ΔM (orange) mice treated with PBS (Untreated), LPS for the indicated periods, or LPS + LPS. A representative macrophage from a 4–5 mice pool for each genotype and condition at 15,000X and 40,000X magnification. Scale bar: 2 µm. **f** At least 6 pictures were quantified for each genotype and condition. (*n* = 7,6,6,8,9,10,6,8,10,7) **g** Heatmap of autophagy-related genes in peritoneal macrophages from *Zeb1*^WT and *Zeb1*^ΔM mice treated with LPS or LPS + LPS. **h** *Sqstm1* mRNA in peritoneal macrophages from *Zeb1*^WT and *Zeb1*^ΔM untreated, treated with LPS or LPS + LPS. (*n* = 5,4,6,4,4,4). **i** As in Fig. 2c, but for p62/SQSTM1 and β-ACTIN. Blots are representative of four independent experiments. **j** Ultrastructure of autophagic vacuoles in LPS-treated macrophages. TEM images of *Zeb1*^WT and *Zeb1*^ΔM macrophages from mice treated with LPS for 3 h. Yellow asterisks: autophagic vacuoles in *Zeb1*^WT macrophages, two with increased electron density (autolysosomes). N nucleus, M mitochondria, ER endoplasmic reticulum. Scale bar: 1000 nm. A representative macrophage from a 4–5 mice pool for each genotype and condition. At least 6 pictures were analyzed for each genotype and condition. Graph bars in Fig. 2 represent mean values +/− SEM with two-tailed unpaired Mann–Whitney test. $p \leq 0.001$ (***), $p \leq 0.01$ (**) or $p \leq 0.05$ (*) levels, or non-significant (ns) for values of $p > 0.05$. Raw data along $p$ values for statistical analyses are included in the Source Data file.

mitochondria undergoing mitophagy (Fig. 2j and Supplementary Fig. S2G). These data suggest that ZEB1 promotes autophagy in the context of acute inflammation.

## Metformin depends on ZEB1 expression in macrophages for its anti-inflammatory effects

Prompted by the above data suggesting that the decrease in mitochondria in macrophages under the LPS + LPS condition accounts for their compromised immune response, we investigated the effects of Metformin, known for inhibiting mitochondrial function and the inflammatory response of macrophages to LPS[17]. We hypothesized that the anti-inflammatory effects of Metformin mimic the immunosuppression observed in macrophages

under the LPS + LPS condition. To test that hypothesis, we examined the in vivo and in vitro effects of Metformin in the response to LPS.

Mouse and human macrophages were treated in vitro with a single dose of LPS in the presence or absence of Metformin or with two doses of LPS (Fig. 3a and Supplementary Methods). As expected, Metformin reverted the LPS-induced upregulation of *IL6* in both human and mouse macrophages (Supplementary Fig. S3A and S3B). In human macrophages, pre-treatment with Metformin before LPS resulted in upregulation of ZEB1 and reduced expression of MT-CO1, which mirrored the expression changes observed in the immunosuppressed PBMCs of septic patients at 72 h shown above or following two doses of LPS (Supplementary Figs. S3C and S3D).

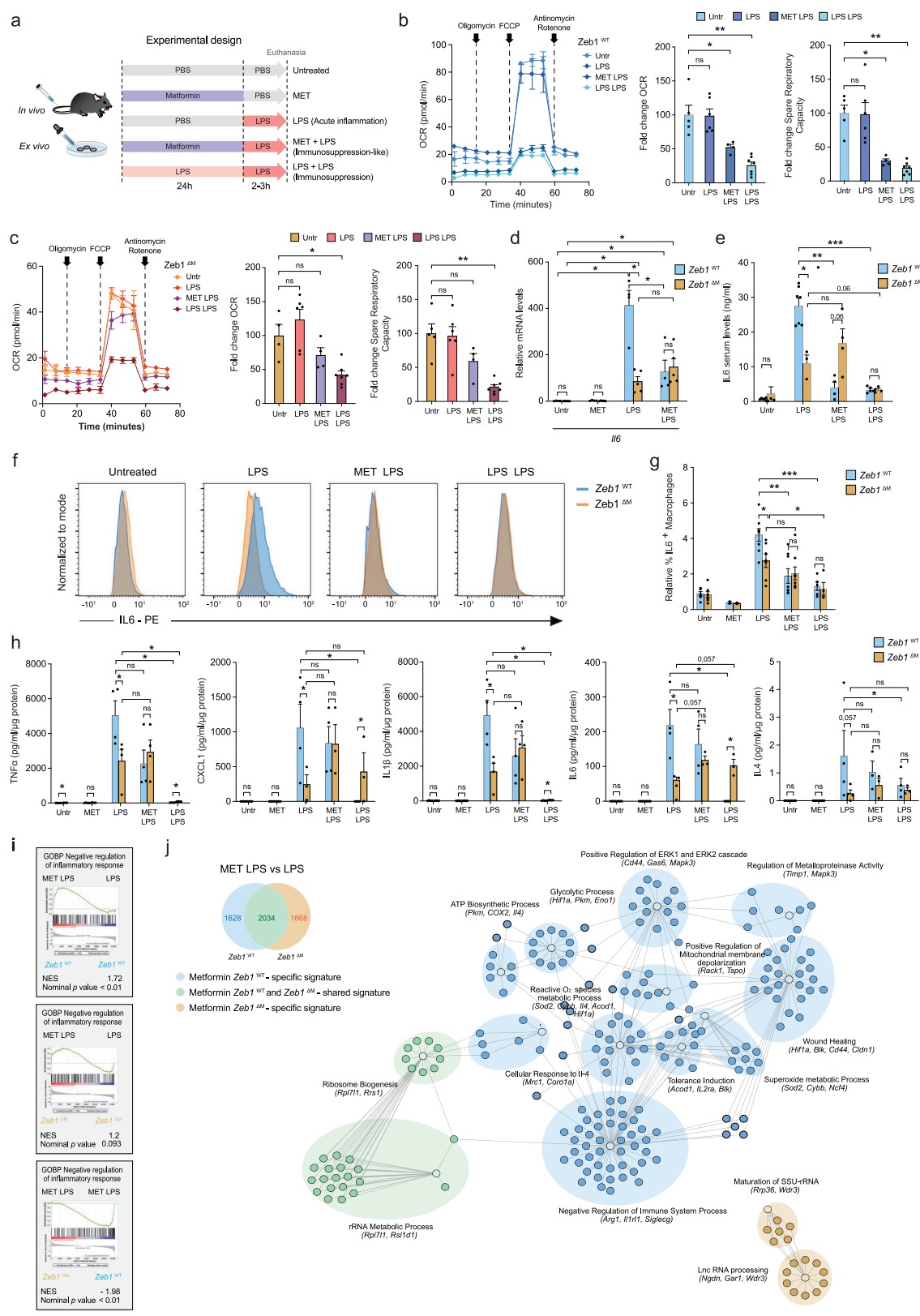

Metformin reduces the oxygen consumption rate (OCR) of macrophages treated with LPS[21]. To determine whether ZEB1 modulates the effect of Metformin on the overall metabolic profile of macrophages during inflammation and immunosuppression, *Zeb1*[WT] and *Zeb1*[ΔM] macrophages were subjected to the protocols in Fig. 3a and assessed for their OCR by Seahorse cell metabolic flux analysis (Fig. 3b, c). We found that the pre-treatment of *Zeb1*[WT] macrophages

with Metformin prior to LPS resulted in a reduction in basal OCR and Spare respiratory capacity to levels comparable to those found in macrophages treated with LPS + LPS. Additionally, Metformin did not alter OCR in *Zeb1*[ΔM] macrophages, suggesting that the metabolic effects of Metformin on OCR require the presence of ZEB1.

To test whether the anti-inflammatory effects of Metformin also depended on ZEB1, peritoneal macrophages from *Zeb1*[WT] and *Zeb1*[ΔM]

**Fig. 3 | Metformin depends on ZEB1 expression in macrophages for its anti-inflammatory effects. a** Experimental design for the in vivo and ex vivo treatment with Metformin (MET). **b** Oxygen consumption rates (OCR) of $Zeb1^{WT}$ macrophages, either untreated or subjected to the LPS, Metformin + LPS or LPS + LPS protocols as in (**a**), were assessed by Seahorse XF Cell Mito Stress Test Kit. Average from at least two independent experiments each including two mice per genotype and condition, each in triplicate. Quantification of basal OCR and Spare Respiratory Capacity of $Zeb1^{WT}$ macrophages ($n = 5,6,4,7$). Untreated is set to 100. **c** As in (**b**), but for $Zeb1^{\Delta M}$ macrophages. **d** $Il6$ mRNA levels in peritoneal macrophages from $Zeb1^{WT}$ and $Zeb1^{\Delta M}$ untreated (PBS), treated with Metformin, LPS or Metformin + LPS ($n = 4,5,4,5,4,5,4,5$). **e** $Zeb1^{WT}$ and $Zeb1^{\Delta M}$ mice were injected i.p. with PBS or LPS, Metformin + LPS or LPS + LPS. IL6 serum levels were measured by ELISA 3 h after the last LPS injection. ($n = 6,4,7,3,4,4,5,4$ mice). **f** Intracellular IL6 was assessed by FACS in F4/80+ peritoneal macrophages from $Zeb1^{WT}$ and $Zeb1^{\Delta M}$ subjected to the indicated treatments. A representative plot of at least three independent experiments. **g** Quantification of IL6+ macrophages in (**f**) ($n = 7,5,2,2,8,7,8,8,5,4$). **h** Cytokine/chemokine production was assessed in the supernatant of peritoneal macrophages from $Zeb1^{WT}$ and $Zeb1^{\Delta M}$ mice subjected to the indicated treatments. $n = 4$ with a pool of two mice per sample. **i** GSEA plots for negative regulation of inflammatory response signature comparing LPS versus Metformin + LPS in $Zeb1^{WT}$ or $Zeb1^{\Delta M}$ macrophages or Metformin + LPS in $Zeb1^{WT}$ versus $Zeb1^{\Delta M}$ macrophages. **j** Venn diagram and gene ontology Cytoscape analysis on the effect of Metformin (MET) in the gene signature of LPS (MET + LPS versus LPS) for each genotype. Specific Metformin signatures for $Zeb1^{WT}$ (blue) $Zeb1^{\Delta M}$ (orange) or shared signatures (green). Each node shows selected DEGs associated with its own GO cluster. Graph bars in Fig. 3 represent mean values +/− SEM with two-tailed unpaired Mann–Whitney test. $p \leq 0.001$ (***), $p \leq 0.01$ (**) or $p \leq 0.05$ (*) levels, or non-significant (ns) for values of $p > 0.05$. Numerical values had been added for $0.05 < p < 0.075$. Raw data along $p$ values for statistical analyses are included in the Source Data file.

mice were subjected to the experimental conditions in Fig. 3a, and their inflammatory status was assessed by their $Il6$ expression. As expected, Metformin alone had no effect on $Il6$ mRNA levels of $Zeb1^{WT}$ and $Zeb1^{\Delta M}$ macrophages (Fig. 3d). However, pre-treatment with Metformin reduced the induction of $Il6$ by LPS in $Zeb1^{WT}$ macrophages but had no effect in LPS-treated $Zeb1^{\Delta M}$ macrophages (Fig. 3d). Furthermore, the effect of Metformin on the systemic inflammatory status in mice of both genotypes was evaluated by measuring their serum levels of IL6. Consistent with the findings in the septic shock model, $Zeb1^{WT}$ mice exhibited higher levels of IL6 in response to LPS compared to $Zeb1^{\Delta M}$ mice (Fig. 3e). Remarkably, Metformin effectively inhibited the LPS-induced upregulation of IL6 in $Zeb1^{WT}$ mice, while no significant effect was observed in $Zeb1^{\Delta M}$ mice (Fig. 3e). Similar results were observed in the analysis of IL6 intracellular staining by FACS in macrophages of both genotypes (Fig. 3f, g).

To gain a comprehensive understanding of the inflammatory response of $Zeb1^{WT}$ and $Zeb1^{\Delta M}$ macrophages, we evaluated the protein levels of a panel of inflammatory markers using a quantitative bead-based cytokine/chemokine multiplex array, which was analyzed by FACS (Fig. 3h). As expected, Metformin alone did not have any effect on macrophages from either genotype. Under the LPS condition, $Zeb1^{\Delta M}$ macrophages exhibited reduced levels of IL1β, IL6, TNFα, and CXCL1 compared to LPS-treated $Zeb1^{WT}$ macrophages. Interestingly, the pretreatment with Metformin brought the expression levels of these inflammatory markers in $Zeb1^{\Delta M}$ macrophages to a similar level as that of $Zeb1^{WT}$ macrophages (Fig. 3h). In contrast, under the LPS + LPS condition, $Zeb1^{\Delta M}$ macrophages exhibited higher levels of IL6 and CXCL1, whereas no detectable levels of IL1β or TNFα were produced by macrophages of either genotype (Fig. 3h). These findings suggest that the anti-inflammatory effects of Metformin in macrophages are dependent on and, at least partially, mediated by ZEB1.

Comparing our RNAseq data with a published RNAseq (GSE98731) of Metformin-regulated genes in alveolar macrophages during an inflammatory response to air pollution in wild-type mice[22], we found that like the alveolar macrophages in the aforementioned study, our $Zeb1^{WT}$ peritoneal macrophages, in contrast to $Zeb1^{\Delta M}$ peritoneal macrophages, exhibited enrichment of Metformin-induced genes during LPS-induced immunosuppression (Supplementary Fig. S3E). This led us to explore by RNAseq the gene signature regulated by Metformin in vivo in peritoneal macrophages isolated from $Zeb1^{WT}$ and $Zeb1^{\Delta M}$ mice that had been pre-treated with Metformin before the administration of LPS. Metformin differentially induced a "negative regulation of the inflammatory response" signature in $Zeb1^{WT}$ macrophages relative to $Zeb1^{\Delta M}$ macrophages (Fig. 3i). Of all the DEGs between the conditions LPS and Metformin + LPS, Metformin regulated 2,034 genes in the same direction in $Zeb1^{WT}$ and $Zeb1^{\Delta M}$ macrophages. However, 1,628 and 1,668 genes were specifically regulated by Metformin in $Zeb1^{WT}$ and $Zeb1^{\Delta M}$ macrophages. Interestingly, within gene signature specific for $Zeb1^{WT}$ macrophages, Metformin increased

genes associated with an anti-inflammatory response (e.g., $Arg1, Mrc1, Il4$) and gene annotations related to the induction of tolerance (e.g., $Acod1$), positive regulation of mitochondrial depolarization (e.g., $Tspo$), wound healing, metalloproteinase activity (e.g. $Timp1, Cldn1$), reactive oxygen species (ROS) metabolic processes (e.g., $Sod2$). In turn, in the specific signature of $Zeb1^{\Delta M}$ macrophages, Metformin-regulated genes were associated with a signature of maturation of SSU-rRNA (e.g., $Wdr3$) (Fig. 3j). These analyses suggested that dependence on ZEB1 for Metformin's anti-inflammatory effects is at least in part due to the regulation of mitochondrial function.

## ZEB1 mediates the immunosuppression-mimicking effect of Metformin by reducing mitochondrial content and ROS levels

The late stages of an acute systemic inflammatory response are characterized by the apoptosis of the majority of immune cells, which contributes to the subsequent immunosuppression stage[43]. The accumulation of ROS and oxidative stress trigger apoptosis, while autophagy serves as an adaptive mechanism to counteract oxidative stress to overcome apoptosis.

At the end of the LPS protocol, $Zeb1^{\Delta M}$ mice exhibited a decreased macrophage count compared to $Zeb1^{WT}$ mice (Supplementary Fig. S4A). $Zeb1^{\Delta M}$ macrophages exhibited higher levels of apoptosis during acute inflammation as assessed by Annexin V (Supplementary Fig. S4B) and a lower expression of anti-apoptotic genes (e.g., $Bcl2$ and $Mcl1$) (Supplementary Fig. S4C).

During acute inflammation, there is a reduction in ATP generation from mitochondria, leading to an increase in the membrane potential ($\Delta\Psi m$), which is necessary for the generation of ROS[9]. In response to LPS, $Zeb1^{WT}$ macrophages exhibited lower ATP levels compared to $Zeb1^{\Delta M}$ macrophages (Supplementary Fig. S4D). In contrast, pre-treatment with Metformin or exposure to LPS + LPS resulted in the opposite effect. In macrophages from $Zeb1^{WT}$ mice, treatment with Metformin + LPS or LPS + LPS resulted in the upregulation of an anti-oxidant signature—as many of the genes in the GSEA annotation "superoxide metabolic process"—compared to $Zeb1^{WT}$ mice treated with LPS alone. This signature was also upregulated in $Zeb1^{WT}$ mice treated with Metformin + LPS or LPS + LPS compared to $Zeb1^{\Delta M}$ mice with the same treatments.

In order to assess the in vivo antioxidant effect of pre-treatment with a previous dose of LPS, we measured ROS production using the luminescence probe L-012 (8-amino-5-chloro-7-phenyl-pyrido[3,4-d]pyridazine-1,4(2H,3H)dione). It was found that $Zeb1^{WT}$ mice exhibited higher levels of ROS compared to $Zeb1^{\Delta M}$ mice during treatment with LPS, but not during LPS + LPS (Supplementary Fig. S4E). Furthermore, we also evaluated ROS production, mitochondrial content, and $\Delta\Psi m$ of macrophages from $Zeb1^{WT}$ and $Zeb1^{\Delta M}$ mice during LPS and LPS + LPS using staining with 6-carboxy-2′,7′-dichlorodihydrofluorescein diacetate ($CH_2$-DCFDA), MTG, and tetramethylrhodamine methyl ester perchlorate (TMRM), respectively. In line with the transcriptomic

analysis shown in Fig. 4a, in LPS, pre-treatment with Metformin or a first sublethal dose of LPS resulted in reduced ROS production, mitochondrial content, and ΔΨm in *Zeb1*[WT] macrophages (Fig. 4b–d). These effects were not observed in *Zeb1*[ΔM] macrophages under the same treatment. Additionally, Metformin pre-treatment decreased TMRM and $CH_2$-DCFDA staining in human macrophages treated with LPS (Supplementary Fig. S4F and S4G). Similar reductions in MTG and TMRM staining were observed in CD14[+] PBMCs from septic patients at 72 h compared to baseline levels at 0 h (Fig. 4e).

Based on the above results, we can draw three main conclusions. Firstly, *Zeb1*[WT] macrophages under acute inflammation exhibited higher ROS production compared to *Zeb1*[ΔM] macrophages. Secondly, *Zeb1*[ΔM] macrophages under immunosuppression showed higher mitochondrial content, ΔΨm, and ROS production compared to *Zeb1*[WT] macrophages. Lastly, Metformin pre-treatment reduced mitochondrial content, ΔΨm, and ROS levels in *Zeb1*[WT] macrophages, while no such reduction was observed in *Zeb1*[ΔM] macrophages.

### ZEB1 inhibits mitochondrial protein translation in inflammatory macrophages

Metformin pre-treatment resulted in a reduction of phosphorylated p65 (P-p65) and MT-CO1 protein expression in inflammatory human macrophages (LPS) to levels comparable to those found in immunosuppressed human macrophages (LPS + LPS) (Fig. 4f, g, and Supplementary Fig. S4H). Analysis of our RNAseq data showed that *Zeb1*[ΔM] macrophages treated with LPS had higher expression of a signature related to "positive regulation of mitochondrial translation" compared to *Zeb1*[WT] macrophages (Fig. 4h). To explore the potential involvement of ZEB1 in mitochondrial translation in macrophages under LPS treatment, we examined the incorporation and tracing of L-homo-propargylglycine (HPG), an analogue of methionine that enables the quantification of newly synthesized proteins. We used emetine to specifically inhibit cytosolic translation and doxycycline to inhibit mitochondrial translation[44,45].

As expected, the combination of doxycycline and emetine more effectively reduced the incorporation of HPG (as a measure of new protein synthesis) in *Zeb1*[WT] macrophages compared to emetine alone (Supplementary Fig. S4I). While there was no difference in mitochondrial protein synthesis between *Zeb1*[WT] and *Zeb1*[ΔM] macrophages under basal conditions, treatment of mice from both genotypes with LPS or LPS + LPS resulted in higher mitochondrial translation in *Zeb1*[ΔM] macrophages compared to *Zeb1*[WT] macrophages (Fig. 4i). These findings suggest that ZEB1 modulates the macrophage response during acute inflammation and immunosuppression by influencing mitochondrial translation.

Inhibition of mitochondrial mRNA translation by tetracyclines reduces in vitro LPS-induced macrophage upregulation of inflammatory cytokines and ameliorates lung and liver damage in endotoxin-induced systemic inflammation[11]. We found here that doxycycline inhibition of ROS production and *Il6* expression in inflammatory macrophages treated is also dependent on ZEB1 expression; in contrast to *Zeb1*[WT] macrophages, pre-treatment with doxycycline did not have an effect in LPS-treated *Zeb1*[ΔM] macrophages (Fig. 4j, k, and Supplementary Fig. S4J). In human monocyte-derived macrophages, doxycycline exhibited a similar immunosuppressive effect as LPS + LPS, resulting in the reduction of MT-CO1 expression Supplementary Fig. S4K).

### Lactate and the inhibition of mitochondrial translation inhibit chronic inflammation in a psoriasis model

We questioned whether the role and mechanism of action of ZEB1 in the regulation of acute inflammation and immunosuppression are conserved in the context of chronic autoimmune inflammation. To that effect, we selected the psoriatic disease, an immune-mediated chronic inflammatory condition affecting not only the skin but also other tissues like joints, where myeloid cell metabolism plays a key pathogenic role[13,46]. While psoriasis is primarily characterized by erythematous and indurated skin plaques, it is considered a systemic inflammatory disease that is associated with increased levels of inflammatory markers in the serum, and about a third of patients will develop psoriatic arthritis (PsA)[47,48].

We investigated whether elevated lactate levels or decreased mitochondrial translation can limit chronic inflammation, similar to their effects in acute inflammation. Topical application of imiquimod (IMQ), a TLR7/8 activator, on the ear of mice leads to the development of psoriasiform skin lesions and epidermal thickening (acanthosis), resembling milder forms of human psoriatic skin lesions[49,50].

The ears of *Zeb1*[WT] mice were left untreated or treated with imiquimod, with or without prior and concurrent systemic treatment of PBS, lactate, or doxycycline (Fig. 5a). Doxycycline and lactate exert their effects through different mechanisms. Doxycycline reduces inflammation by suppressing mitochondrial translation, while lactate exerts its effects by triggering anti-inflammatory and reparative responses[10,11]. We explored whether inhibition of mitochondrial translation by doxycycline has an anti-inflammatory effect not only in acute inflammation but also in chronic inflammation. Treatment with lactate and doxycycline in *Zeb1*[WT] mice, improved psoriasiform lesions, resulting in reduced ear and epidermal thickening (Fig. 5b, c), and increased lactate production by their macrophages (Fig. 5d). Doxycycline increased ZEB1 expression while reducing MT-CO1 protein levels, whereas lactate had the opposite effects (Fig. 5e and Supplementary Fig. S5A and S5B). Additionally, compared to lactate, doxycycline exhibited greater effectiveness in reducing systemic inflammation, as evidenced by its effects on imiquimod-induced splenomegaly and ROS production (Supplementary Fig. S5C and S5D).

### Metformin requires ZEB1 expression in macrophages for its anti-inflammatory effects in psoriasis

We then investigated whether the reliance on ZEB1 for the anti-inflammatory effect of Metformin observed in the context of acute inflammation also applies to psoriasis. imiquimod treatment resulted in reduced *Zeb1* expression in the ear sections of *Zeb1*[WT] mice (Supplementary Fig. S5E), and it led to greater erythema and epidermal thickening in the ears of *Zeb1*[ΔM] mice compared to *Zeb1*[WT] mice (Fig. 5f, g). IMQ-treated *Zeb1*[ΔM] mice also exhibited greater systemic inflammation than *Zeb1*[WT] mice as evidenced by the larger spleens in the former (Supplementary Fig. S5F). Pre-treatment of mice with Metformin prior to imiquimod resulted in a reduction of macroscopic skin lesions, histological acanthosis, and macrophage infiltration in *Zeb1*[WT] mice, while it had no effects in *Zeb1*[ΔM] mice (Fig. 5f–h). These findings further support the dependence of Metformin's anti-inflammatory effects on the expression of ZEB1 in macrophages.

ZEB1 was found in scattered cells in the healthy skin but it was nearly absent in psoriatic skin lesions (Supplementary Fig. S5G). As reported, ZEB1 is upregulated in melanoma skin lesions[51] (Supplementary Fig. S5G). We identified ZEB1 in CD68[+] macrophages in healthy skin that are negative for ZEB1 in the psoriatic skin (Fig. 5i and Supplementary Fig. S5I). In addition, our analysis of a published gene microarray (GSE14905)[52] also revealed the downregulation of *ZEB1* in psoriatic skin lesions (Supplementary Fig. S5H). ZEB1 was also expressed in CD68[+] macrophages in the synovial membrane of PsA patients but it was nearly absent in the synovium of osteoarthritis patients (Fig. 5i and Supplementary Fig. S5I). We also found that in contrast to the immunosuppressed PBMCs of septic patients at 72 h, the monocyte-enriched PBMCs of PsA patients exhibited higher mitochondrial content and ROS production compared to PBMCs from healthy donors (Fig. 5j, k, and Supplementary Fig. S5J). As in endotoxin-induced immunosuppression (LPS + LPS), ZEB1

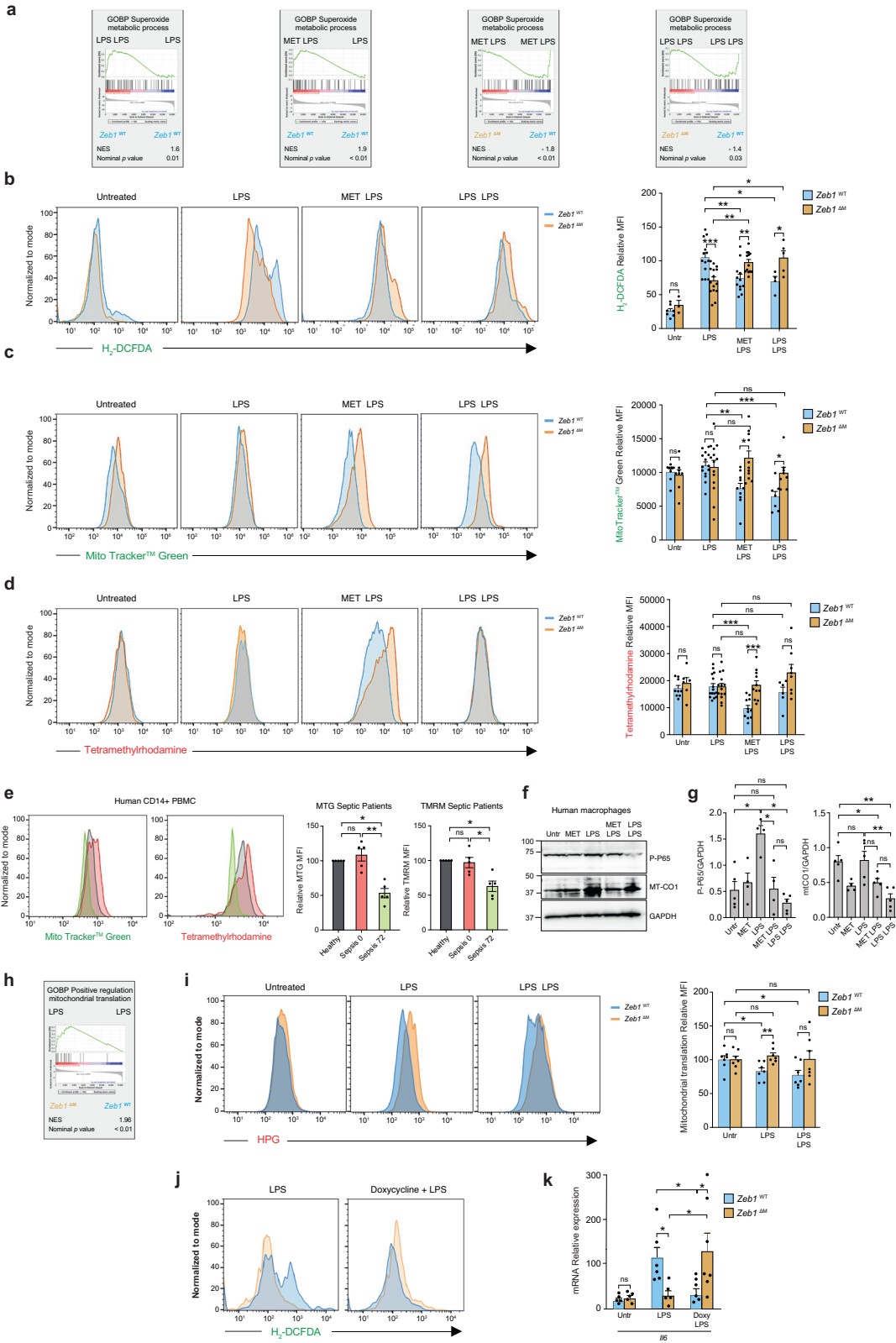

expression by macrophages is also necessary to restrict chronic inflammation and Metformin was dependent on ZEB1 expression for its anti-inflammatory role.

In summary, our findings suggest that ZEB1 expression in macrophages is required for limiting both acute and chronic inflammation, as well as for the anti-inflammatory effects of Metformin in both forms of inflammation.

## ZEB1 inhibits mitochondrial protein translation by restricting amino acids transport

After showing that ZEB1 regulates the inflammatory and immunosuppressive responses of macrophages through inhibiting mitochondrial translation, we sought to further investigate the underlying molecular mechanism. Analysis of our RNAseq indicated that $Zeb1^{\Delta M}$ macrophages expressed higher levels of gene signatures associated with

**Fig. 4 | ZEB1 mediates Metformin's immunosuppression-mimicking effect by reducing mitochondrial content and ROS levels. a** GSEA plots for "Superoxide metabolic process" signature comparing macrophages from $Zeb1^{WT}$ and $Zeb1^{\Delta M}$ mice subjected to the indicated treatments. **b** $Zeb1^{WT}$ and $Zeb1^{\Delta M}$ mice were injected i.p. with PBS, LPS, Metformin + LPS, or LPS + LPS and ROS production was assessed by FACS for CH$_2$-DCFDA staining in F4/80+ cells. A representative plot of 6–15 mice per genotype and condition in four independent experiments. CH$_2$-DCFDA MFI in macrophages ($n = 7,3,13,15,12,13,4,5$ mice). **c** As in (**b**), macrophages were assessed for MTG staining ($n = 7,7,13,13,9,10,6,7$ mice). **d** As in (**b**), macrophages were assessed for TMRM ($n = 9,6,15,14,11,11,6,8$ mice). **e** At the indicated time, human CD14$^+$ PBMCs from a healthy donor and septic patients were assessed for MTG and TMRM staining. Representative FACS plots ($n = 5$ individuals per condition) and MFI quantification. **f** A representative blot of at least four independent experiments to assess MT-CO1 and P-p65 in human macrophages treated as indicated. **g** Quantification of MT-CO1 ($n = 5,4,5,5,5$) and P-P65 expression relative to GAPDH ($n = 5,4,4,4,5$) in all experiments for (**f**). **h** GSEA plot for "Positive regulation of mitochondrial translation" annotation comparing macrophages from LPS-treated $Zeb1^{WT}$ and $Zeb1^{\Delta M}$ mice. **i** Mitochondrial translation in macrophages from $Zeb1^{WT}$ and $Zeb1^{\Delta M}$ mice either untreated or treated with LPS or LPS + LPS was assessed by FACS for L-HPG with Alexa Fluor® 647 picolyl azide. Representative plots of $n = 7$ independent experiments. Quantification analysis of all experiments. **j** CH$_2$-DCFDA staining of macrophages from $Zeb1^{WT}$ and $Zeb1^{\Delta M}$ mice treated with LPS or doxycycline + LPS. The FACS plots shown are representative of a total of 4–5 mice per genotype and condition assessed in two independent experiments. **k** $Il6$ mRNA in macrophages from $Zeb1^{WT}$ and $Zeb1^{\Delta M}$ mice either untreated or treated with LPS or doxycycline + LPS. Average of 5–6 mice for each genotype assessed in two independent experiments ($n = 4,5,5,5,6,7$). Graph bars in Fig. 4 represent mean values +/− SEM with two-tailed unpaired Mann–Whitney test. $p \le 0.001$ (***), $p \le 0.01$ (**) or $p \le 0.05$ (*) levels, or non-significant (ns) for values of $p > 0.05$. Raw data along $p$ values for statistical analyses are included in the Source Data file.

mitochondrial tRNA aminoacylation and tRNA modification (Supplementary Fig. S6A). The methylthiotransferase $Cdk5rap1$ catalyzes 2-methyl-2-thio (ms2) modifications of several mt-tRNAs and its deficiency hampers mitochondrial protein synthesis and OxPhos activity and increases ROS production[53]. LPS treatment resulted in the downregulation of $Cdk5rap1$ expression in $Zeb1^{WT}$ macrophages but not in $Zeb1^{\Delta M}$ macrophages (Supplementary Fig. S6B). In contrast, ms2 tRNA modifications were reduced in $Zeb1^{\Delta M}$ macrophages after one dose of LPS but increased after two doses of LPS (Supplementary Fig. S6C). Given its opposing effects on the regulation of $Cdk5rap1$ expression and ms2 tRNA modifications, we concluded that ZEB1 inhibits mitochondrial protein translation through mechanisms independent of these processes.

Both cytoplasmic and mitochondrial protein translation, as well as autophagy, are regulated by mTOR, which is activated by the uptake and metabolism of amino acids[54,55]. Analysis of our RNAseq revealed that when $Zeb1^{WT}$ macrophages were pre-treated with Metformin (MET + LPS) or exposed to a previous dose of LPS (LPS + LPS), there was a decrease in the expression of GSEA annotations associated with amino acid transport (Fig. 6a). Furthermore, $Zeb1^{WT}$ macrophages in the LPS + LPS condition exhibited reduced levels of an amino acid transport signature compared to $Zeb1^{\Delta M}$ macrophages (Supplementary Fig. S6D). The L-type bidirectional amino acid transporter SLC7A8/LAT2 is required for the uptake of glutamine and branched-chain amino acids (BCAA), which are critical for the activation of mTORC1 signaling[56,57]. Treatment of $Zeb1^{WT}$ macrophages with Metformin + LPS and LPS + LPS led to a decrease in the mRNA and protein levels of SLC7A8 compared to LPS treatment, whereas this effect was not observed in $Zeb1^{\Delta M}$ macrophages (Fig. 6b, c, Supplementary Fig. S6E). In line with the upregulation of SLC7A8 during LPS, we observed that $Zeb1^{WT}$ macrophages but not $Zeb1^{\Delta M}$ macrophages displayed higher glucose and glutamine consumption specifically in the LPS condition (Fig. 6d, e). We then aimed to explore the potential alterations in intracellular levels and consumption of BCAAs as well as glutamine and its derived amino acid glutamate in inflammatory macrophages. We observed that, similar to glucose and glutamine, a single dose of LPS resulted in increased levels of BCAAs in $Zeb1^{WT}$ macrophages, but not in $Zeb1^{\Delta M}$ macrophages (Fig. 6f). Conversely, BCAAs levels were comparable between $Zeb1^{WT}$ and $Zeb1^{\Delta M}$ macrophages treated with Metformin + LPS and LPS + LPS. Furthermore, we found a reduction in the intracellular levels of glutamate in LPS-treated $Zeb1^{\Delta M}$ macrophages compared to their $Zeb1^{WT}$ counterparts. Of note, no significant difference was observed in intracellular glutamine levels between LPS-treated $Zeb1^{\Delta M}$ and $Zeb1^{WT}$ macrophages, suggesting that glutamine may have already been metabolized to glutamate at the analyzed time point (Fig. 6g).

We hypothesized that if ZEB1 inhibits mitochondrial translation by downregulating SLC7A8 expression, restricting glutamine availability would alleviate this effect. To test this, we treated $Zeb1^{WT}$ and $Zeb1^{\Delta M}$ macrophages with LPS or LPS + LPS in the presence or absence of glutamine. In the absence of glutamine, there was a decrease in MTG staining in both genotypes and across all treatments. Consequently, the decrease in mitochondrial content observed in $Zeb1^{WT}$ macrophages treated with LPS + LPS compared to $Zeb1^{\Delta M}$ macrophages, which was observed in the presence of glutamine, was prevented in the glutamine-free condition (Supplementary Fig. S6F). These results suggest that ZEB1 plays a role in restricting amino acid consumption, which is essential for mTORC1 activation and mitochondrial translation.

We also assessed mitochondrial DNA (mtDNA) copy number (MDCN) in $Zeb1^{WT}$ and $Zeb1^{\Delta M}$ macrophages under our different experimental conditions and in the presence or absence of glutamine. In the presence of glutamine and compared to a single dose of LPS, pre-treatment with Metformin (Metformin + LPS) or treatment with LPS + LPS reduced MDCN in $Zeb1^{WT}$ macrophages but not in $Zeb1^{\Delta M}$ macrophages (Fig. 6h). However, in the absence of glutamine, MDCN was comparable between LPS-treated $Zeb1^{WT}$ and $Zeb1^{\Delta M}$ macrophages. Additionally, the treatments of Metformin + LPS or LPS + LPS did not alter MDCN in either $Zeb1^{WT}$ and $Zeb1^{\Delta M}$ macrophages compared to LPS treatment. Taken together, the above data suggest that in the presence of glutamine, treatment with Metformin or LPS before adding a second dose of LPS (Metformin + LPS or LPS + LPS) results in a reduction in mitochondrial content and MDCN in LPS-treated $Zeb1^{WT}$ macrophages, but not in $Zeb1^{\Delta M}$ macrophages. Based on these findings, two conclusions can be drawn: (1) MDCN in inflammation (LPS) is dependent on the availability of glutamine, and (2) glutamine levels in immunosuppressed (LPS + LPS as well as Metformin + LPS) macrophages are regulated by ZEB1 expression.

## ZEB1 inhibits mTORC1/p70S6K signaling in immunosuppression

mTORC1 serves as an energy sensor with pleiotropic functions, including the regulation of nuclear DNA-encoded mitochondrial protein translation[44]. Activation of mTORC1 is primarily driven by growth factors and nutrient availability, particularly amino acids[56,57]. Regulation of protein synthesis by mTORC1 is mediated through its phosphorylation and activation of p70 ribosomal protein S6 kinase (p70S6K, encoded by $Rps6kb1$)[44]. Given the inhibitory effect of Metformin on mTORC1 signaling[58], we aimed to investigate whether the mechanism by which ZEB1 regulates immunosuppression and mediates the anti-inflammatory effects of Metformin is through the reduction of amino acid uptake and metabolism, leading to the downregulation of mTORC1.

Compared to $Zeb1^{WT}$ and $Zeb1^{\Delta M}$ macrophages treated with a single dose of LPS, Metformin pre-treatment (MET + LPS) or a subsequent dose of LPS (LPS + LPS) resulted in reduced P-p70S6K levels in $Zeb1^{WT}$ macrophages, but not in $Zeb1^{\Delta M}$ macrophages (Fig. 6i and Supplementary Fig. S6G). By examining the levels of phosphorylated p70S6K (P-p70S6K), IL6, and SLC7A8 in PBMCs from healthy controls, septic patients at 0 h (immunogenic), and septic patients at 72 h

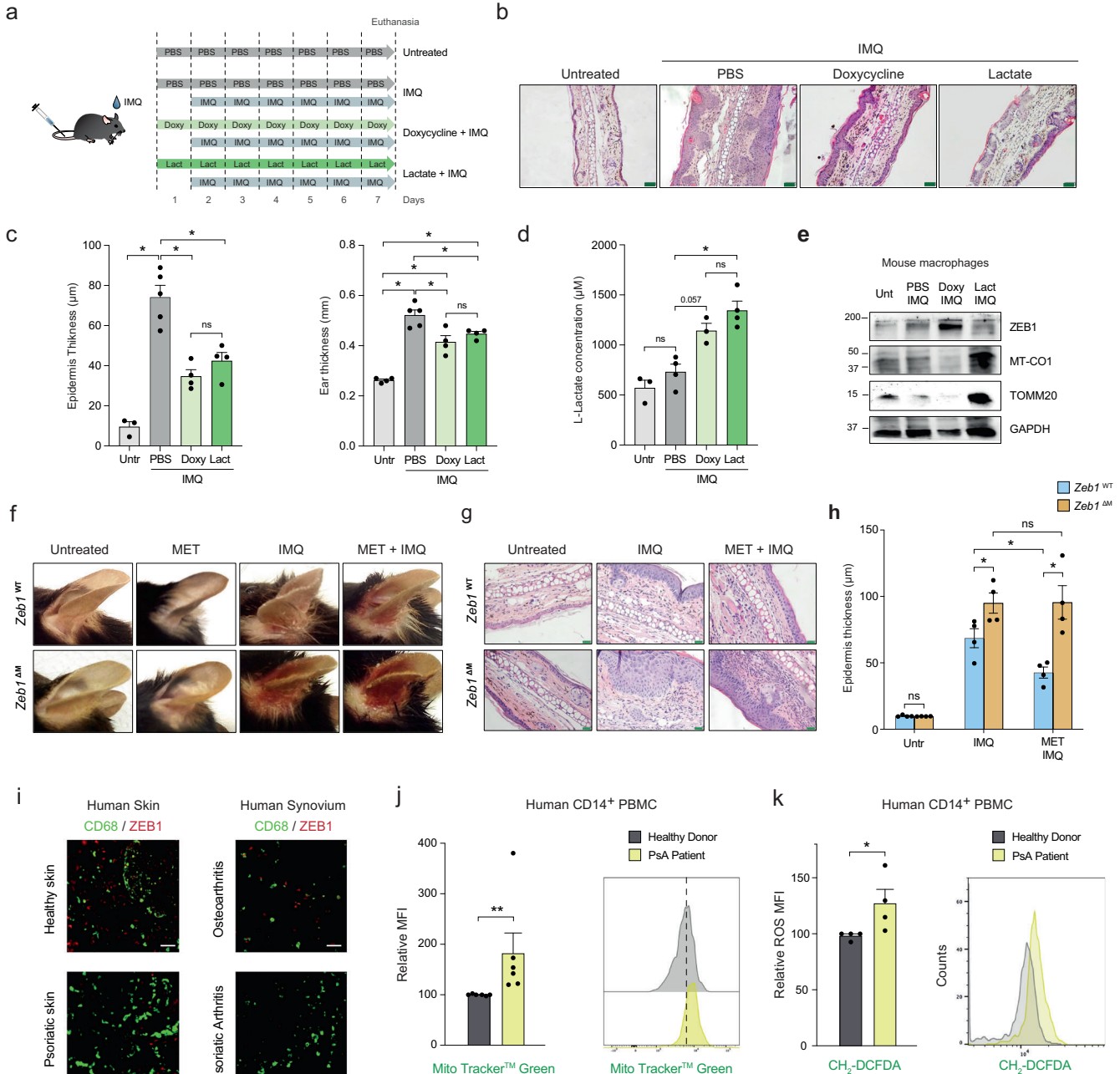

**Fig. 5 | Metformin requires ZEB1 expression in macrophages for its anti-inflammatory effects in psoriatic disease lesions. a** Schematic of the protocols used in the Imiquimod (IMQ) mouse model. Ten-to-twelve weeks age male mice treated with Imiquimod and either PBS, doxycycline, or lactate. **b** H&E staining of histological ear sections of mice either untreated and treated with Imiquimod, Lactate + Imiquimod, or Doxycycline + Imiquimod. Scale bar, 50 μm. **c** Quantification of the epidermal ear thickness assessed in four pictures from at least four mice per condition as in (**b**) ($n = 3, 5, 4, 4$ mice for each condition). Quantification of the ear thickness of four mice per condition. **d** Lactate levels in the supernatant of peritoneal macrophages from wild-type mice left untreated or treated with PBS + Imiquimod, Doxycycline + Imiquimod, or Lactate + Imiquimod ($n = 3, 4, 3, 4$). **e** Western blot for ZEB1, MT-CO1, TOMM20 and GAPDH in peritoneal wild-type macrophages treated as in (**b**). **f** Representative pictures from three independent experiments of *Zeb1*^WT and *Zeb1*^ΔM mice after 7 days of treatment with PBS, Metformin, Imiquimod or Metformin +

Imiquimod. **g** As in (**f**), but ear sections were counterstained for H&E. Scale bar: 50 μm. **h** As in (**f**), epidermal ear thickness in four mice per genotype and condition in three independent experiments was quantified by ImageJ. Four separate areas in two pictures were quantified for each mouse ear ($n = 4$). **I** CD68 and ZEB1 staining along with DAPI in the skin samples from healthy donors and psoriatic patients as well as synovial membrane samples from osteoarthritis and PsA patients. Representative captions of at least two independent experiments. Scale bar: 25 μm. **j** MFI quantification and representative FACS plots for MTG staining in CD14+ PBMCs from six PsA patients relative to the MTG's MFI of six healthy controls ($n = 6$). **k** As in (**j**), but CH₂-DCFDA staining in CD14⁺ PBMCs from four healthy donors and four PsA patients ($n = 4$). Graph bars in Fig. 4 represent mean values +/− SEM with two-tailed unpaired Mann–Whitney test. $p ≤ 0.001$ (***), $p ≤ 0.01$ (**) or $p ≤ 0.05$ (*) levels, or non-significant (ns) for values of $p > 0.05$. Numerical values had been added for $0.05 < p < 0.075$. Raw data along $p$ values for statistical analyses are included in the Source Data file.

(immunosuppressed), we observed decreased levels of all three in the PBMCs of sepsis 72 h, indicating an immunosuppressive state (Fig. 6j and Supplementary Fig. S6H and S6I). In contrast to the down-regulation of MT-CO1 in PBMCs from septic patients at 72 h, the PBMCs

from PsA patients exhibited an upregulation of MT-CO1 (Supplementary Fig. S6H). Additionally, PBMCs from septic patients at 72 h showed reduced mitochondrial respiratory capacity compared to healthy donors (Supplementary Fig. S6J). Collectively, these results suggest

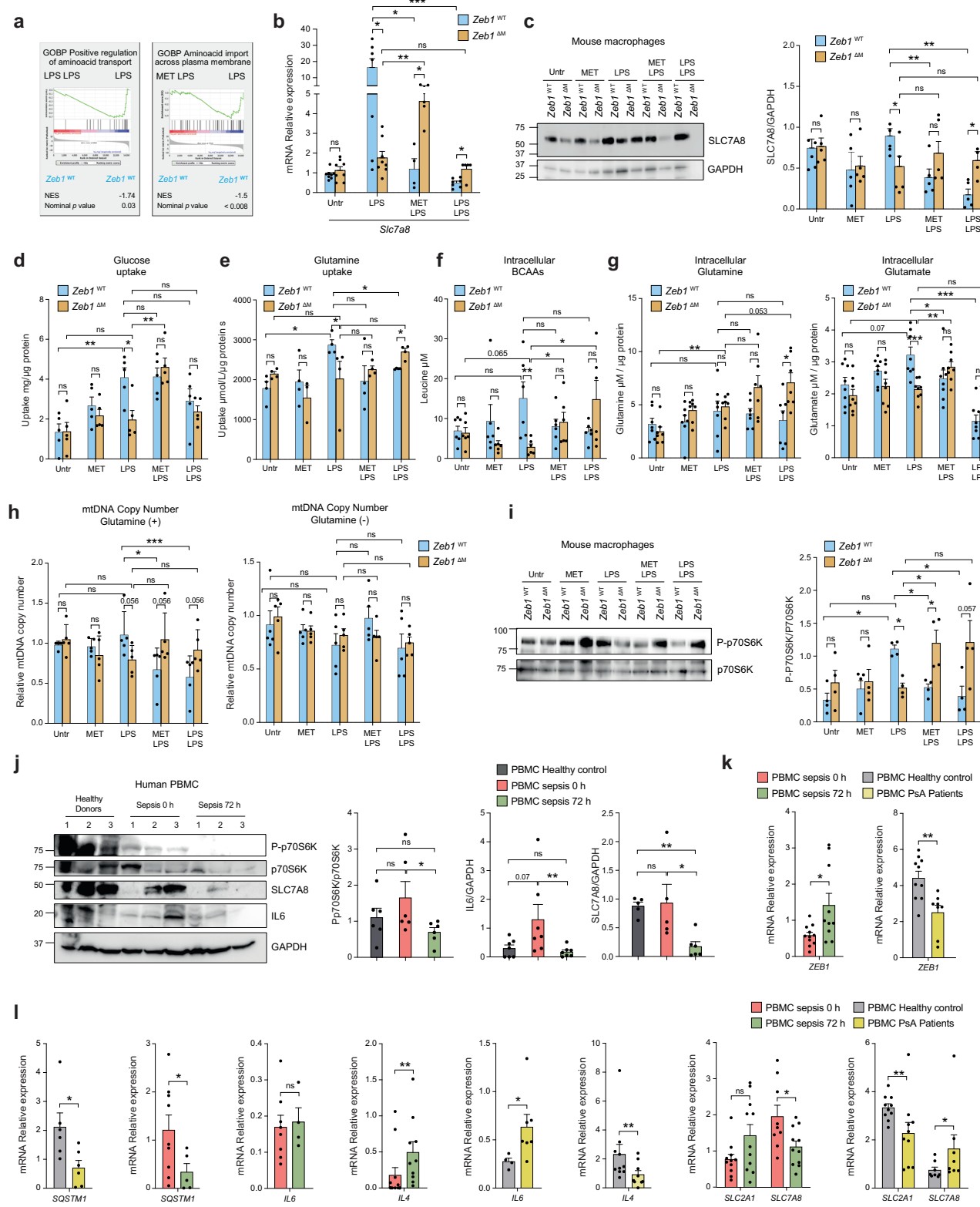

that ZEB1 expression is required for immunosuppression and Metformin's inhibition of mTORC1.

**ZEB1 expression in inflammatory PBMCs from septic and PsA patients is associated with differential expression of glucose and amino acid transporters**

Next, we examined whether the expression of ZEB1 in PBMCs from healthy controls, septic patients at 0 h and 72 h, and PsA patients

correlated with the expression of several markers in our sepsis and psoriasis mouse models. In septic patients, the expression levels of *ZEB1* were higher in PBMCs at 72 h compared to PBMCs at 0 h. On the other hand, in PBMCs from PsA patients, *ZEB1* expression was lower when compared to PBMCs from healthy donors (Fig. 6k). The PBMCs from septic patients at 0 h exhibited higher expression of *SQSTM1* and reduced expression of *IL4* compared to the PBMCs from the same patients at 72 h (Fig. 6l). In contrast, and compared to PBMCs from

**Fig. 6 | ZEB1 inhibits mitochondrial protein translation by restricting amino acid transport. a** GSEA plot "Amino acid transport" signature in *Zeb1*[WT] macrophages under the indicated treatments. **b** *Slc7a8* mRNA in macrophages from *Zeb1*[WT] and *Zeb1*[ΔM] mice subjected to the indicated treatments. Average of 4–10 mice per genotype and condition in three independent experiments ($n = 8,8,7,8,4,5,7,5$). **c** Representative blot of five independent experiments for SLC7A8 and GAPDH in peritoneal macrophages under the indicated treatments. Quantification of SLC7A8 relative to GAPDH ($n = 5$). **d** Glucose uptake by peritoneal macrophages from *Zeb1*[WT] and *Zeb1*[ΔM] treated as indicated and assessed 4 h after the last LPS dose ($n = 5$). **e** As in (**d**), but for glutamine uptake ($n = 4$). **f** As in (**d**), but for intracellular BCAAs in cell lysates 4 h after the last LPS dose ($n = 6$). **g** As in (**f**), but for intracellular glutamine ($n = 7$) and glutamate ($n = 8$) levels corrected by cell protein levels. **h** Relative mtDNA copy number in macrophages from *Zeb1*[WT] and *Zeb1*[ΔM] mice treated as indicated and cultured with (left panel) or without (right panel) glutamine in the medium ($n = 5$). **i** Western blots for p70S6K, P-P70S6K, and GAPDH in *Zeb1*[WT] and *Zeb1*[ΔM] peritoneal macrophages treated as indicated. A representative blot and quantification of P-p70s6k normalized to p70s6k ($n = 4$). (**j**) Western blots for for P-p70s6k, P70S6K, IL6, SLC7A8, and GAPDH in PBMCs of three healthy donors and three septic patients at 0 and 72 h. Representative blot of two independent experiments, with 2-3 patients per condition. Quantification of P-p70s6k protein levels normalized to p70s6k ($n = 6$), as well as IL6 ($n = 7$) and SLC7A8 ($n = 5$) normalized to GAPDH. **k** *ZEB1* mRNA in PBMCs from septic patients at 0 h ($n = 11$) and 72 h ($n = 10$), healthy donors ($n = 10$), and PsA patients ($n = 8$). **l** As in (**k**), but for *SQSTM1* in septic patients ($n = 9,5$) and Healthy/PsA ($n = 6$), *IL4* in septic patients ($n = 12,10$) and Healthy/PsA ($n = 10,9$), *IL6* in septic patients ($n = 8,4$) and Healthy/PsA ($n = 4,7$), *SLC2A1* in septic patients ($n = 11$) and Healthy/PsA ($n = 10$), and *SLC7A8* in septic patients ($n = 9,11$) and Healthy/PsA ($n = 8$). Graph bars in Fig. 6 represent mean values +/− SEM with two-tailed unpaired Mann–Whitney test. $p \leq 0.001$ (***), $p \leq 0.01$ (**) or $p \leq 0.05$ (*) levels, or non-significant (ns) for values of $p > 0.05$. Raw data along $p$ values for statistical analyses are included in the Source Data file.

healthy donors, the PBMCs from PsA patients showed lower expression of *SQSTM1* and *IL4* but higher of *IL6* (Fig. 6l).

A reverse pattern of expression was observed for glucose transporter *SLC2A1* and amino acid transporter *SLC7A8* between septic and PsA patients. PBMCs from septic patients at 0 h exhibited lower levels of *SLC2A1* and higher levels of *SLC7A8* compared to PBMCs from septic patients at 72 h (Fig. 6l). Additionally, PBMCs from PsA patients exhibited lower levels of *SLC2A1* and higher levels of *SLC7A8* compared to PBMCs from healthy donors (Fig. 6l).

Psoriatic human skin exhibited upregulation of both MT-CO1 and P-p65 compared to healthy donor skin (Supplementary Fig. S6K). Consistent with these findings, analysis of the published array GSE57383[59] revealed a similar reverse pattern of expression for *ZEB1*, *SLC2A1*, and *SLC7A8* in the myeloid CD14[+] PBMCs of healthy donors and PsA patients (Supplementary Fig. S6L). Taken together, these findings support a model where ZEB1 modulates the inflammatory phenotype of human PBMCs in sepsis and psoriatic disease through amino acid efflux-dependent regulation of mitochondrial translation.

Finally, we investigated the association of *ZEB1* expression with sepsis outcomes by analyzing a published array (GSE48080) of patients with sepsis caused by community-acquired pneumonia[60]. The PBMCs from patients who eventually did not survive have higher levels of *ZEB1* and *TNF*, and lower levels of *IL4* at the time of diagnosis (day 0), compared to PBMCs from survivor patients (Supplementary Fig. S6M). Seven days after the diagnosis, the PBMCs from non-survivors still have higher levels of *ZEB1*, but they exhibit the opposite pattern of *TNF* and *IL4* expression compared to day 0 (Supplementary Fig. S6N). Accordingly, there is a positive correlation between *ZEB1* and *TNF* expression at day 0, but a negative correlation at day 7 (Supplementary Fig. S6N). On the other hand, while there was no correlation between *ZEB1* and *IL4* expression at day 0, there was a positive correlation between the two genes at day 7 (Supplementary Fig. S6N).

## Discussion

Dissecting the mechanisms that regulate the resolution of acute inflammation, the induction of immunosuppression, or the progression to chronic inflammation is essential for designing therapeutic approaches to modulate excessive inflammation or restore immune competence. The present study found that ZEB1 plays a dual role in macrophages, being required developing an inflammatory phenotype but also to limit and resolve inflammation by promoting the transition of macrophages to an immunosuppressive state (schematic in Fig. 7). ZEB1 expression in macrophages was also required for the anti-inflammatory and ROS-inhibiting effect of Metformin in both an endotoxin-induced model of acute inflammation as well as in a chronic inflammation model such as psoriasis. We showed that the reprogramming of both human and mouse macrophages from an inflammatory state to an immunosuppressive state is driven by autophagy and a decrease in mitochondrial translation, which are dependent on their expression of ZEB1. During the acute inflammatory response, ZEB1 promotes glucose and amino acid consumption and upregulates pro-inflammatory and glycolytic genes. Simultaneously, ZEB1 induces autophagy to eliminate damaged mitochondria, and other organelles, thereby preventing excessive inflammation that could lead to cell and tissue damage or contribute to the development of chronic inflammation. Conversely, in an immunosuppressed state, ZEB1 enhances lactate production and the expression of anti-inflammatory genes, while inhibiting amino acid uptake.

Inflammation is a protective response to infection, stress, and injury[4]. However, when it becomes dysregulated, it can result in tissue damage, systemic disease, and even death. On the other hand, immune tolerance serves as a self-regulatory mechanism to safeguard the organism against the detrimental effects of excessive inflammation. Nevertheless, a prolonged immunoparalysis renders the organism incapable of responding to subsequent antigenic challenges. Therefore, the timing and extent of the inflammatory and immunosuppressive responses must be tightly regulated. Here, we found that ZEB1, a regulator of cell plasticity in cancer stem cells, exhibits contrasting roles in inflammation and immunosuppression. This duality makes ZEB1 a particularly suitable factor for regulating macrophage plasticity during both the inflammatory response and its subsequent resolution. Beyond macrophages, ZEB1 plays important functions in other immune cells; it modulates early B cell differentiation and is critical for the maintenance of memory CD8[+] T cells ([61,62], and reviewed in[63]).

Metformin has been tested in preclinical models of several chronic inflammatory autoimmune diseases[15]. It plays pleiotropic roles in immunometabolism, exerting these functions through multiple mechanisms, including inhibiting ETC-CI, activating AMPK, and inhibiting the NRPL3 inflammasome[11,16,17,21,22]. Our results indicate that ZEB1 expression in macrophages is required for the in vivo anti-inflammatory effects of Metformin in response to LPS and in a mouse model of psoriasis. Analysis of our RNAseq of macrophages from *Zeb1*[WT] and *Zeb1*[ΔM] mice treated with Metformin + LPS evidenced that ZEB1-regulated DEGs include genes/pathways involved in antioxidant response, mitochondrial function and amino acid transporters. mTORC1 acts as an energy sensor and plays key functions in cell metabolism and proliferation being activated by growth factors and amino acids (reviewed in[64]). In this study, it was found that ZEB1 inhibits amino acid transport in response to a secondary inflammatory stimulus, thereby reducing mTORC1 signaling and potentially mediating the known inhibitory effect of Metformin on mTORC1 signaling. Interestingly, mTORC1 inhibits EMT and ZEB1 expression in cancer cells[65], suggesting the existence of a loop between mTORC1 and ZEB1, which may regulate cell growth and inflammation.

We found that Metformin reduced mitochondrial content, ΔΨm, and ROS production in LPS-treated *Zeb1*[WT] macrophages to the same

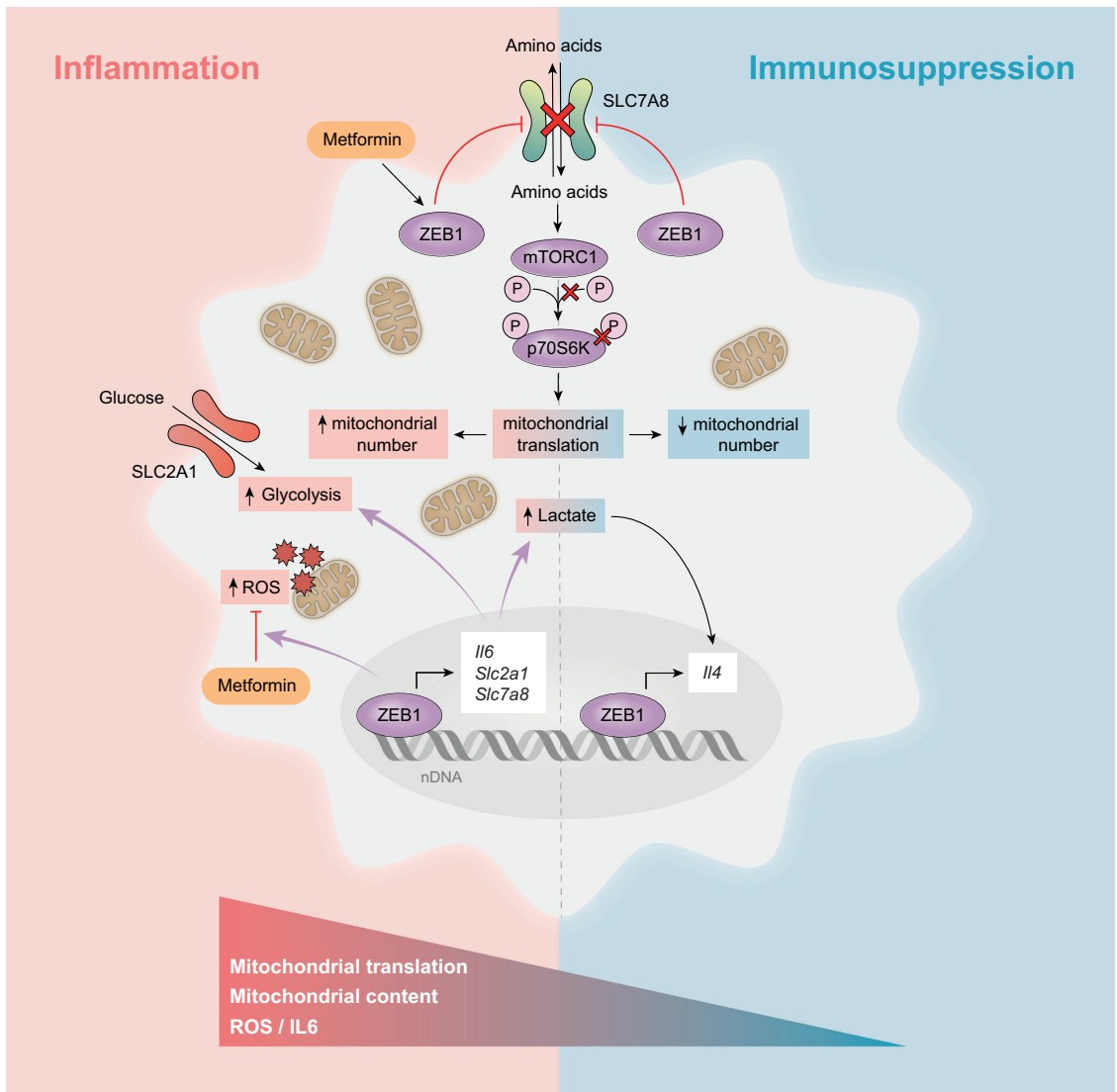

**Fig. 7 | Schematic summary model.** ZEB1 regulates inflammation and immunosuppression in macrophages and is required for the anti-inflammatory and ROS-inhibiting effects of Metformin.

levels as those observed in LPS + LPS macrophages. Although the mechanism by which ZEB1 mediates the effect of Metformin to promote an immunosuppressive status in macrophages remains to be elucidated, Metformin inhibits mTORC1, whose genetic and pharmacological ablation upregulates ZEB1 in cancer cells[5,58,59,66]. We found that relative to LPS, treatment of mouse macrophages with Metformin + LPS or LPS + LPS reduced the phosphorylation of p70S6K as in the immunosuppressed PBMC of septic patients. ZEB1 suppressed mTORC1 activation by inhibiting amino acid transport, which resulted in reduced mitochondrial translation and ROS production, as well as lower inflammatory cytokine production. Interestingly, we found that the role of ZEB1 mediating the anti-inflammatory and ROS-inhibiting effects of Metformin were independent of Metformin's ability to increase glucose uptake. Our results suggest that Metformin is dependent on ZEB1 for the regulation of amino acid transport. However, we can not rule out that ZEB1 may also be involved in other pathways contributing to the anti-inflammatory effects of Metformin, such as autophagy or AMPK signaling.

In contrast to its pro-inflammatory effects in sepsis, ZEB1 had a protective anti-inflammatory role in psoriasis. Its expression was downregulated in the psoriatic disease macrophages. The anti-inflammatory role of Metformin in the IMQ model also depended on

the expression of ZEB1. Inhibition of mitochondrial mRNA translation by tetracycline family antibiotics inhibits LPS-induced production of inflammatory cytokines by macrophages and ameliorates lung and liver damage in endotoxin-induced systemic inflammation[11]. Although the mechanism by which the inhibition of mitochondrial translation ameliorates inflammation is not known, it is likely to involve alterations in the mitochondrial electron transport chain and ROS production. It has been reported that Metformin, by blocking mitochondrial complex I, increases the levels of lactate, which in turn promotes anti-inflammation through lactylation of histones in anti-inflammatory genes.[17,20]. We found here that lactate and the inhibition of mitochondrial translation with doxycycline reverted ROS production and the inflammatory effects of imiquimod. Therefore, we propose a model in which ZEB1 triggers and self-limits the inflammatory responses of macrophages by modulating their metabolism. ZEB1 expression in macrophages increases lactate levels, triggering a homeostatic response through histone lactylation in anti-inflammatory and reparative genes. Additionally, both ZEB1 and metformin reduce macrophage amino acid levels and consumption, thereby inhibiting mTORC1 activity, essential for mitochondrial translation.

Overall, our results uncovered a mechanism regulating the inflammatory and tolerogenic responses of macrophages and set ZEB1

as a potential target in acute and chronic inflammation to prevent hyperinflammation and immunoparalysis.

## Methods

### Human samples

The use of human samples in this study was approved by the local Ethics Research Committee under protocols HCB/2017/0767, HCB/2019/1012, HCB/2020/0100, had the informed consent of patients and conformed with the principles of the Helsinki Declaration. Skin samples from psoriatic and melanoma patients, synovial membranes from psoriatic arthritis and osteoarthritis patients, and peripheral blood from healthy donors, septic patients, and patients with psoriatic arthritis were obtained as detailed in Supplementary Methods.

### Mouse models and isolation of mouse macrophages

The use of mice in this work followed the guidelines established by the Animal Experimental Committee at the University of Barcelona School of Medicine (Barcelona, Spain) and by the Generalitat de Catalonia that reviewed and approved under references 396/18 and 133/19 and 1041, respectively. The *Zeb1*$^{fl/fl}$ mouse (herein referred to as *Zeb1*$^{WT}$) was generated by CRISPR as detailed in Supplementary Methods. The *Zeb1*$^{WT}$ mouse was then crossed with a mouse carrying the Cre recombinase selectively in myeloid cells under the control of the endogenous lysozyme 2 (*Lyz2*, also referred as *LysM*) promoter/enhancer (official name: B6.129P2-*Lyz2*$^{tm1(cre)Ifo}$/J), (The Jackson Labs, Bar Harbor, ME, USA), to generate the myeloid conditional *Zeb1* knockout (*Zeb1*$^{fl/fl}$/*LysM*$^{Cre}$, referred in the manuscript as *Zeb1*$^{ΔM}$) mice. The mice were housed in a temperature-controlled barrier room maintained at 21-22 °C with a 12-hour light/dark cycle. They were provided with standard rodent chow (RM1-P, SDS, Dietex, Argenteuil, France) and had access to water ad libitum. All mice were euthanized by cervical dislocation. The setting of the LPS- and imiquimod-induced models of sepsis, and psoriasis, respectively, as well as the isolation of macrophages, are detailed in the Supplementary Information.

### Determination of mRNA expression and tRNA modifications, and RNA sequencing

mRNA expression and analyses of tRNA modifications were determined by quantitative real-time PCR (qRT-PCR) as described in Supplementary Methods. RNA sequencing was conducted as described in Supplementary Methods.

### Determination of protein expression by FACS, Western blot, and ELISA

Determination of cell surface and intracellular protein expression by FACS and Western blot and of cytokines/chemokines by an enzyme-linked immunosorbent assay or using a beads-based multiplex array are detailed in Supplementary Methods.

### Determination of lactate and amino acids

The intracellular levels of glutamine, glutamate and BCAAs and the uptake of glucose and glutamine were assessed as detailed in Supplementary Methods.

### Assessment of mitochondrial content, ROS and ATP production, mitochondrial membrane potential, lysosome/mitophagy, mitochondrial protein translation, and mtDNA copy number

These parameters were assessed in vitro and/or in vivo as detailed in Supplementary Methods.

### Immunostaining and electron transmission microscopy

Immunostaining and morphological analysis of macrophages by optical and electron transmission microscopy were conducted as in Supplementary Methods.

### Extracellular flux and high-resolution respirometry

Analyses of oxygen consumption and extracellular acidification were assessed in a Seahorse XF$^e$96 Extracellular Flux Analyzer. High-resolution respirometry was carried out in an Oroboros Oxygraph-2k. See Supplementary Methods for details.

### Statistics and reproducibility

All replicates in this study are biologically independent human samples, mice, and peritoneal macrophages. No prior sample size calculation was performed and sample sizes in the experiments were set based on our previous experience and similar studies in the literature. All experimental results were included in the figures, encompassing all data points without any selection or exclusion. Each experiment was independently repeated at least twice with similar results. Blinding was not technically feasible in this study because transgenic mice were cohoused, and therefore, they had to be marked for identification. For all in vivo experiments, after being genotyped, age- and sex-matched mice were distributed in roughly equal numbers to the different treatments. To minimize cage effects, mice of different genotypes and under different treatment conditions were housed together in mixed cages. For in vitro experiments, macrophages from age- and sex-matched mice of each genotype were isolated and randomly assigned to either treatment or control groups. The RNAseq was performed at an external facility and staff were blinded to the genotype or treatment condition of the samples, which were identified by a code. Most data collected and analyzed in the study were quantitative in nature rather than qualitative in nature. Except for RNA-seq experiments, statistical analysis of the data was conducted using Prism for Mac 9.3.1 (GraphPad Software, La Jolla, California). Bar graphs throughout the manuscript represent the mean with standard errors in which the statistical significance was assessed with a non-parametric Mann-Whitney U test. Survival curves in Kaplan Meier plots were compared by the Log-rank (Mantel-Cox) test. Where appropriate, relevant comparisons were labeled as either significant at the $p ≤ 0.001$ (***), $p ≤ 0.01$ (**) or $p ≤ 0.05$ (*) levels, or non-significant for values of $p > 0.05$, and with specified numerical values for $0.05 < p < 0.075$. All raw data along $p$ values for statistical analyses are included in the Source Data file.

### Reporting summary

Further information on research design is available in the Nature Portfolio Reporting Summary linked to this article.

## Data availability

The RNA-seq data have been uploaded to the Gene Expression Omnibus (GEO) database and assigned accession number GSE207328. All relevant data are available in the Source data file, which is provided with this article.

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

## Acknowledgements

We are grateful to Dr. F. Sanchez-Madrid (Hospital Princesa and CNIC, Madrid, Spain), Dr. D. Cebrian (CNIC, Madrid, Spain), and Dr. A. Valledor (University of Barcelona, Spain) for helpful insights on early versions of the manuscript. We thank Dr. C. Stephan-Otto Attolini (BIST-IRB, Barcelona, Spain) and Dr. J. Rios (IDIBAPS, Hospital Clinic, and Autonomous University of Barcelona, Barcelona, Spain) for their expert guidance on the statistical analyses of the data in the study. We also thank Dr. L. Ribas de Pouplana (BIST-IRB, Barcelona, Spain) for advice on mitochondrial translation experiments. We acknowledge technical assistance by staff in the Flow Cytometry Unit at IDIBAPS, the Molecular Interactions Services Unit at the Biomedical Research Institute of Bellvitge (IDIBELL), and the Transmission Electron Microscopy Unit at the University of Barcelona School of Medicine. We also thank A Téllez (Hospital Clinic, Barcelona, Spain) for his help in collecting samples from septic patients, and Dr. MJ Fernández-Aceñero (Hospital Clinico San Carlos, Madrid, Spain) for help in collecting skin samples from healthy controls, psoriatic patients, and melanoma patients. We are also thankful to Dr. DC Dean (University of Louisville, KY, USA) for his generous gift of an anti-ZEB1 polyclonal antibody. We thank Dr. A. Garcia for the artistic drawing of schematics in the article. IDIBAPS is partly funded by the CERCA Programme of Generalitat de Catalunya. The study was conducted at IDIBAPS' Centre de Recerca Biomèdica Cellex building, which was partly funded by the Cellex Foundation. The different parts of this study were independently funded by grants to AP from the Leo Foundation (LF-OC-19-000166), the Catalan Agency for Management of University and Research Grants (AGAUR) (2017-SGR-1174 and 2021-SGR-01328), and the Spanish State Research Agency (AEI) of the Ministry of Science and Innovation (MICINN) (PID2020-116338RB-I00) as part of MICINN's National Scientific and Technical Research and Innovation 2021-2023 Plan, which is co-financed by the European Regional Development Fund (ERDF) of the European Union Commission. AB is a recipient of a PhD scholarship from AGAUR (FI Program, 2021 FI_B 00514).

## Author contributions

M.C. and A.P. conceived, designed and interpreted experiments. M.C. also performed most of the experimental work in the study. A.B., M.C.M.-C. and C.N. conducted some experiments in the study. V.D. and B.P. generated the *Zeb1*^WT mouse. S.F. and P.C. assisted us in procuring human samples from septic patients. R.C. and J.D.C. assisted us in procuring human samples from PsA patients. A.E.-C. conducted bioinformatics analyses of the raw data from the RNAseq. J.J.L. and J.S. performed the Cytoscape analysis. G.G. and A.M.S. provided advice on metabolic experiments. C.E. contributed in the capture and analysis of TEM pictures. MMR conducted the liquid chromatography–mass spectrometry analysis of glutamine levels. A.P. also supervised and obtained funding for the study and wrote the manuscript. All authors provided critical comments to the manuscript.

## Competing interests

The authors declare no competing interests.

## Additional information

[1]Group of Gene Regulation in Stem Cells, Cell Plasticity, Differentiation, and Cancer, IDIBAPS, 08036 Barcelona, Spain. [2]National Center of Biotechnology (CSIC-CNB) and Center for Molecular Biology Severo Ochoa (CSIC/UAM-CBMSO) Transgenesis Facility, Higher Research Council (CSIC) and Autonomous University of Madrid (UAM), Cantoblanco, 28049 Madrid, Spain. [3]Medical Intensive Care Unit and Department of Internal Medicine, Hospital Clínic of Barcelona, Group of Muscle Research and Mitochondrial Function, IDIBAPS, and CIBERER, 08036 Barcelona, Spain. [4]Arthritis Unit, Dept. of Rheumathology, Hospital Clínic and IDIBAPS, 08036 Barcelona, Spain. [5]National Center for Genomics Analysis (CNAG), 08028 Barcelona, Spain. [6]Biomedical Research Networking Centers in Digestive and Hepatic Diseases (CIBERehd), Carlos III Health Institute, 08036 Barcelona, Spain. [7]MRC Metabolic Diseases Unit, University of Cambridge Metabolic Research Laboratories, Wellcome Trust-MRC Institute of Metabolic Science, Addenbrooke's Hospital, Cambridge CB1 0QQ, UK. [8]Department of Biomedicine, University of Barcelona School of Medicine and Health Sciences, 08036 Barcelona, Spain. [9]Department of Biochemistry and Molecular Genetics, Hospital Clínic of Barcelona and IDIBAPS, 08036 Barcelona, Spain. [10]Molecular Targets Program, Division of Oncology, Department of Medicine, J.G. Brown Cancer Center, Louisville, KY 40202, USA. [11]ICREA, 08010 Barcelona, Spain. [12]These authors contributed equally: Agnese Brischetto, M. C. Martinez-Campanario. ✉e-mail: mcortesh@recerca.clinic.cat; idib412@recerca.clinic.cat

