## [Peer Review File · Nature Communications]

Macrophage reprogramming toward an immunosuppressive phenotype is driven by reduced amino acid-dependent mitochondrial translationREVIEWER COMMENTS

Reviewer #1 (Remarks to the Author):

The authors describe that metformin inhibits transcription of inflammatory genes and formation of ROS in macrophages via ZEB1. Moreover, ZEB1 has opposing roles in the induction of inflammation and depression in LPS tolerance via decreasing mitochondrial content, inhibiting protein translation downregulating mTORC1 signaling and induction of autophagy. Metformin mimics ZEB1-reprogramming of macrophages.

Overall, the paper presents an interesting concept of ZEB1 as a switch in macrophage programming, and metformin acting through it.

Yet, there are some important problems that need to be addressed.

First, a number of controls are missing, e.g. metformin alone both in vitro and in vivo, vehicle injection in mice.

Second, the manuscript is hard to follow, which is in part due to the complicated tolerance-inducing schemes with and without inhibitors, and in part due to poor annotation, very small fonts, incomplete legends, breaking up of figures in too many subfigures (instead of focusing on the meaningful and organizing accordingly). etc.

Third, the number of repetitions changes from condition to condition in a series of experiments without explanation and contrary to standards. This raises questions about selection of data and omission of data points.

Fourth, in many cases single measurements are depicted, and neither quantification nor statistical analysis are performed

All of these issues make the paper appear immature.

Further important problems following the sequence of figures.

Figure 1.

A) The legend is insufficient. Exact numbers of mice have to be provided for each genotype and each condition separately. Furthermore, it needs to be stated in detail, how many mice died due to the intervention, and how often mice had to be euthanized. Any mouse that was not analyzed in full (i.e. was taken out of the analysis) has to be depicted. Please elaborate: What was your reason to use this number of mice, i.e. was there a predefined number. All of this information is essential!

B) The authors define peritoneal macrophages as CD45+ CD11b+ F4/80+. It would be desirable to see the gating strategies of all conditions: untreated, LPS, LPS+LPS. As there is probably beginning influx of inflammatory monocytes 24h post LPS treatment, we recommend to add Ly6C to the panel to rule out that there is monocyte contamination in their "peritoneal macrophages" if they compare them to the LPS 3h condition. Ideally, the authors should provide a panel that shows all myeloid cells that could express LysM (neutrophils, monocytes, dendritic cells), with respect to the later experiments.

B-D) It seems that Zeb1ko macrophages are non-responsive with regard to LPS stimulation (either one or two). It would be interesting to see transcripts, if present immunity related, that are actually upregulated under the Zeb1 knock out conditions. The depiction should be simplified by combining 1B and 1D. Instead, levels of multiple cytokines, not just IL6 should be measured e.g. by bead assays to complement the transcriptional analysis.

C + H) All western blots need to be analyzed by densitometry (including statistics of n=3 or more samples).

E) The authors show a panel of transcripts that are involved in glycolysis. It would be nice if this panel could be completed, i.e. add remaining enzymes such as Gapdh, Pgk1, Eno2, Pkm, Ldha.

E) - F) The authors suggest that decreased levels of lactate play a role in the impaired anti-inflammatory transition of Zeb1 knock out macrophages. They show the transcript of monocarboxylic acid transporter Slc16a member 1, however the member 3 (Slc16a3) might have a higher affinity for lactate and should be analyzed.

I) Here, and in all instances comparisons of all (!) data point of one experiment need to be statistically analyzed. The selection of comparisons seems random, this needs to be changed.

J) Could the authors please provide a set of genes that are associated with autophagy/mitophagy or autophagy of mitochondria (similar to Figure S1K). This would relieve the impression of choosing only differentially expressed genes.

Figure S1

G) Please indicate conditions of the western plot.

- J) Please indicate conditions of the western plot (right).
- K) Please show color code of the heatmap.
- K) To be able to fully grasp the differences observed in mitophagy under these conditions, the authors should provide a set of genes associated with autophagy of mitochondria (, e.g. GO term 0000422; which includes Usp30 and Sqstm1). Also why are the chosen genes not analyzed in the Zeb1 knock out condition? This might add a piece to the puzzle.
- L) Please show color code of the heatmap.
- L) Why does the macrophage induce mitochondria DNA-encoded gene transcription, i.e. increases the production of mitochondria content, to shortly after (~27h; Figure 1 G) remove mitochondria from the cell via autophagy? Assuming that the decrease of mitochondria numbers would also happen without a second LPS administration. Could the authors please speculate.
- M) Please indicate conditions of the western plot

Figure 2.

- A) Is not referenced in the manuscript. Instead of Figure 2A, 2B is referenced.
- B) Should be referenced a sentence later.
- C) Analysis as above.
- D) E) What are the authors referring to as „OCR“ in the right graph? It would be helpful if the authors could also indicate the measured spare capacity in the figure or figure legend. How many cells did the authors use, as basal respiration seems quite low. The authors refer to ECAR in that section however, they do not show ECAR plots or quantification. I suggest to remove ECAR from the text.
- F) The following controls are missing in vivo and need to be provided: The vehicle control for metformin and LPS, and metformin alone. These controls are very important (injections are a treatment themselves). These data are very important as a basis for the transcriptome analysis in Fig. 2i.
- Line 292, referring to Figure 1A, instead it should refer to Figure 1B
- G) Again, essential controls are missing (see 2F), in this case even Zeb1ΔMac untreated and LPS only condition. Accordingly, „thus, eliminating the ZEB1-mediated inflammatory difference between both genotypes showed above in Figure 1B (Fig. 2G) is overstated given the fact that controls are missing and „only“ IL-6 transcript is measured compared to an inflammatory gene set in Figure 1B.

Figure S2.

- A) Please indicate conditions of the western plot
- Overall: Single measurements are not acceptable. Quantification (densitometry) and statistical analysis is absolutely required).

Figure 3.

- A) The GSEA are not in line with what is measured in Figure 3SE. While there is a clear difference in ROS production visible in the in vivo luminescence assay, there is no significant difference in the gene sets of Zeb1wt and Zeb1ΔMac treated with a single dose of LPS. Vice versa, only one of the two representative mice of Zeb1wt in Figure 3E shows a luminescence signal. Suggesting less ROS production in Zeb1wt, what we would expect given the results from Figure 3 B), but opposing a significant difference in the GSEA of 2xLPS treatment in the „favor“ of Zeb1wt in this subfigure. Also, I would advise to only state the genotype underneath the enrichment score of the GSEA and better not add the pre-treatment condition as well. The treatment conditions in the top line are sufficient and „MET LPS in Zeb1ΔMac MET“ is confusing in my opinion.
- B) “Metformin reduced ROS production in Zeb1WT macrophages but not in Zeb1ΔMac ones” This statement is wrong: Metformin increases ROS production in Zeb1ΔMac! Please explain. Furthermore, it is not acceptable that for each in vitro condition a different number of measurements are performed (or just depicted). In all experimental conditions the n has to be similar (at least 3 independent measurement).
- D) Line 357 refers to Figure 3E instead of D)
- E) One patient? Please measure at least 5-10 patients, otherwise this figure has to be deleted.
- F) See E).
- H) In the manuscript the authors refer to a „emetine alone“ condition, but there is no such condition in Figure 3H.

Figure S3.

- D) The authors note that a drop of ATP production is necessary to produce ROS. Accordingly, ATP production in Zeb1wt mice is less than in Zeb1ΔMac after single LPS treatment. However,

compared to the untreated controls the Zeb1wt macrophages remain unchanged and the Zeb1ΔMac macrophages drop in ATP production.

E) To visualize the overall low luminescence signal better, it might help to shave the fur of the mice.

Overall: Single measurements are not acceptable. Quantification (densitometry) and statistical analysis is absolutely required).

Figure 4.

C) Right panel: How do peritoneal macrophages contribute to the lactate formation as a mediator of psoriasis in the skin? Or why did the authors choose to look at peritoneal macrophages? I would suggest to have a new subfigure and do not put that data into the ear histology measurement data.

G) Again, the "metformin only" condition is missing.

I) Again, only one patient, i.e. not acceptable.

Figure S4.

D) E) It would be nice to see a Zeb1ΔMac untreated spleen quantified and as a picture.

Figure 5.

B) Again, highly variable numbers of repetitions are shown for each condition in vitro. This is unacceptable.

E) Why do the authors use bone marrow-derived macrophages in this experiment? All the other experiments were carried out in peritoneal macrophages and glucose and glutamine uptake could also be performed in peritoneal macrophages.

Reviewer #2 (Remarks to the Author):

Review of the paper "Metformin depends on ZEB1 expression in macrophages for its anti-inflammatory effects" by Cortes et al.

The study by Cortes et al., is focused on the role of the transcription factor ZEB1 as a major regulator of both acute macrophage activation and their transition to an immune tolerant/inflammatory resolution state. Using both mouse and human cells, the authors propose a mechanism in which under conditions of acute inflammation, Zeb1 increases glucose metabolism to control inflammatory gene expression. In contrast, under tolerogenic states, such as re-exposure of macrophages to LPS or Psoriasis, the authors describe a scenario in which Zeb1 expression limits amino acid uptake via control of the expression of glucose and amino acid transporters, leading to the suppression of mTORC1 signaling, reduced translation of mitochondrial biogenesis, induction of autophagy, increased lactate production, and the expression of anti-inflammatory genes. Lastly, the authors found that metformin treatment mimicked macrophage tolerance, and Zeb1 was required for the anti-inflammatory effects of metformin.

Overall, we find that study of the role of Zeb1 as a major regulator of acute activation and tolerance in macrophage to be novel and of high importance to the field of innate-immunity and immunometabolism. However, as written and presented the manuscript is convoluted, hard to follow, and the data does not fully support the authors conclusions. A primary critique of the manuscript is the experiments are generally missing key details, some main findings are not statistically significant, relies on superficial analysis to draw main conclusions, and does not use multiple approaches to support or further test key findings. Additionally, we find the manuscript to be confusing to follow, since the writing is not very clear, lacking in many cases conclusions or explanations of the main findings. We find it may help significantly improve the manuscript if the authors increase the quality of the data versus the quantity of the data. For example, increasing the depth of the study by focusing on a few strongly supported mechanisms/pathways vs those that are weakly supported. Please see below for other major and minor concerns:

Major:

- 1) For in vivo studies do the authors see any sex differences in the phenotypes between males and females? Please include the age and sex of all mice used in the study.
- 2) For in vitro studies (including supplemental data), please include cell type used (BMDMs,

peritoneal macrophages, etc.) clear stimulation details such as dose and timing of stimuli.

3) Please properly label all Western blots including those in supplemental such as Figure S1G, S1J, etc.)

4) In figure 1, both in vitro and in vivo experimental conditions (acute and immunosuppression conditions), can the authors carefully measure Zeb1 expression kinetics for both transcript and protein levels? This was done in S1K, only under acute LPS exposure. In S1K we see that acute LPS reduces Zeb1 expression by 24 hours, thus if Zeb1 expression is down-regulated how is it also involved or necessary for LPS tolerance upon re-exposure to LPS?

5) For Fig1 and FigS1, cytokines such as IL-6 should be validated by ELISA.

6) Lines 265-267: "We hypothesized that metformin-mediated inhibition of ETC-I could mimic the immune tolerance in macrophages after two doses of LPS" Why?

7) Figure 2DE-Not consistent with literature. Mouse macrophage shut down OCR after LPS treatment. What are the experimental conditions and why does this data contrast with the well-established finding in the field that LPS inhibits the electron transport chain and OCR?

8) Line 281: Typo, should say "To test whether the in vivo anti-inflammatory effects of metformin depended on Zeb1..."

9) Line 306 refers to Figure 2J, which is not included in the manuscript data figures.

10) Line 325-329. Confusing paragraph. What is the main point? Please clarify.

11) Line 330-332: As stated above, please be clear in the paper that these are peritoneal macrophages.

12) Line 345-351: Instead of saying we validated, we suggest the authors explain how they tested their hypothesis. Otherwise, the experiment sounds biased or predetermined.

13) 361-366: This paragraph is a summary of the observations. However, the authors should provide more insight and the implications of these observations to help the reader better understand what this means in regards to the role of Zeb1 in these processes.

14) Line 371: S3H, MTCO1 is not the same image/exposure cropped in the main figure.

15) Line 380/Figure 3H: There is no emetine alone treatment.

16) Figure 3I/S3I: The difference in mitochondrial protein translation is modest and does not appear to be significant between WT and Zeb1 KO macrophages. Thus, there is no data supporting the conclusion of lines 385-387, which is a major finding the author claim Zeb1 regulates.

17) Figure 3J/S3J: doesn't appear that ROS is statistically significantly between doxy-LPS WT and Zeb1 KO macrophage. Also, representative flow plot used in Fig 3J, does not match the MFI data in Fig S3J.

18) Line 415-419: What's is the conclusion of this data? Please provide insight to what these observations mean.

19) Figure 4 and SF4: IF staining is weak. It is hard to see colocalization of CD68 and ZEB1.

20) Line 440-446: Confusing paragraph, please make clearer.

21) Lines 461-476: It was hard to follow why "ZEB1 inhibits mitochondrial protein translation during inflammation, through mechanisms independent of for CDK5RAP1 and mt-tRNAs modifications". Please make this paragraph clearer and provide more insight and logic how this conclusion was made.

22) Figure 5E: The data only shows that ZEB1 KO macrophages take up less glucose and more glutamine, but the data does not prove they consume or metabolize it.

23) Line 481: Reference 50 does not show that SLC7A8/LAT2 "is required for the efflux of glutamine to incorporate leucine, thus activating mTORC1 signaling". In fact, they argue the opposite.

24) Figure 5F, very modest phenotype. Can the authors confirm SLC7A8 via western blot?

25) Figure 5H. The qPCR data does not match the expression data presented in the flow data in Figure F. For example, LPS treated WT macrophages have significantly more SLC7A8 mRNA transcript compared to LPS treated ZEB1 KO macrophage, but less protein expression. Which means the metformin-LPS treatment qPCR data is not reliable as a measurement of SLC7A8 protein expression.

26) Figure 5G: Besides mitotracker green, mitochondria content could be measured by looking at the expression of electron transport complexes and/or mito DNA copy number.

Minor:

1) Line 167: Insert Figure number 1D at the end of sentence. Also, it's not fully accurate to call these anti-inflammatory genes, MMPs for example can also be pro-inflammatory.

2) Callout for Figure 2B should be Figure 2A (which is missing), and proper callout for Figure 2B

should be at the end of the sentence in line 270.

3) Figure 4D: straighten cropped images.

4) Line 479: typo

5) Line 481: typo

July 7, 2023

RESPONSE TO REFEREES

Below is a summary of general changes introduced in the manuscript plus all comments made by the reviewers (in *italic and bold*) followed by our response.

A) OVERALL CHANGES

- With the inclusion of new data, the revised manuscript now contains 130 figure panels, which is a 25% increase compared to the original manuscript's 104 panels.
- Importantly, to avoid any potential confusion, it should be noted that the data previously presented in former Figures 2, 3, 4, and 5 are now presented in Figures 3, 4, 5, and 6, respectively.
- Additionally, in order to enhance clarity and visual presentation, a professional graphic designer has been commissioned to draw all the main figures and selected Supplementary Figures in the revised manuscript.

B) REPLY TO REVIEWER #1

The authors describe that metformin inhibits transcription of inflammatory genes and formation of ROS in macrophages via ZEB1. Moreover, ZEB1 has opposing roles in the induction of inflammation and depression in LPS tolerance via decreasing mitochondrial content, inhibiting protein translation downregulating mTORC1 signaling and induction of autophagy. Metformin mimics ZEB1-reprogramming of macrophages. Overall, the paper presents an interesting concept of ZEB1 as a switch in macrophage programming, and metformin acting through it.

We appreciate the positive comments of Reviewer #1 regarding the interest of the manuscript.

Yet, there are some important problems that need to be addressed.

First, a number of controls are missing, e.g. metformin alone both in vitro and in vivo, vehicle injection in mice.

We are very grateful to Reviewer #1 for the chance to clarify this important issue. We summarize and in detail response to Reviewer #1's specific comments to the former Figure 2F the explanation of the inclusion of controls in the study.

→ Regarding the control of metformin alone. This issue is explained in detail in our response to Reviewer #1's comments to the former Figure 2F. Although we had already performed the experiments, the "metformin" alone condition was not included in the original manuscript because of two reasons: a) in the absence of LPS, metformin alone has no effects on inflammatory markers and b) as this Reviewer indicates below, some figures (including Supplementary Figure S1) were already too big. In any case, following her/his suggestion, the revised manuscript includes the metformin-alone condition in the first inflammatory *in vitro* and *in vivo* experiments where metformin was used (new Figures 3D, 3G, 3H and 5F).

→ Regarding the injection of the corresponding vehicle of active compounds. This issue is explained in detail in our response to Reviewer #1's comments to the former Figure 2F. "Untreated" mice and mice receiving only one stimulus (metformin alone, IMQ alone, one dose of LPS) were always injected with PBS. The use of PBS as vehicle control was already specified in the protocols included in Supplementary Methods. Again, we would like to note that given the word limit for figure legends (350 words) in Nature Communications, not all this information can be included in the legend and has to be relegated to Supplementary Methods. In any case, following Reviewer #1's suggestion, the use of vehicle has been stressed in the revised manuscript.

Second, the manuscript is hard to follow, which is in part due to the complicated tolerance-inducing schemes with and without inhibitors, and in part due to poor annotation, very small fonts, incomplete legends, breaking up of figures in too many subfigures (instead of focusing on the meaningful and organizing accordingly). etc.

We appreciate Reviewer #1's constructive criticisms regarding the size and display of the figures.

We concur with her/him that the original manuscript contained a large number of Figures and Supplementary Figures. This is because the study was previously reviewed in another journal, whose reviewers requested experiments in multiple and divergent directions. We tried to streamline and consolidate the manuscript when it was transferred to *Nature Communications*.

→ Following this Reviewer's comments, the revised manuscript has implemented the following changes regarding the number and display of figures:

- The revised manuscript has been reorganized throughout to streamline and focus its content on the main points. Some accessory figures have eliminated or transferred to the Supplementary section. For instance, seven figures have been eliminated (Fig. 2G, Fig. 3A, Fig. 5E, Fig. 5F, Suppl Fig. S1K; and Suppl Fig S2B) (see below in this letter for the reasoning) and six have been moved to the Supplementary Section (former Fig. 2B to Suppl Fig S3A, former Fig. 2C to Suppl Fig. S3B, former Fig. 3H to Suppl Fig. S4I, former Fig. 5A to Suppl Fig S6A, former Fig. 5B to Suppl Fig. S6B, and former Fig. 5C to Suppl Fig. 6C).

- As mentioned in the "Overall changes" section, a professional graphic designer has redrawn all the main figures and a selection of Supplementary Figures in the revised manuscript. This redesign incorporates larger fonts and clearer displays for improved readability.
- Within the strict 350-word limit established by *Nature Communications* for each figure legend, all figure legends have been revised to include information previously relegated to Supplementary Methods.

Third, the number of repetitions changes from condition to condition in a series of experiments without explanation and contrary to standards. This raises questions about selection of data and omission of data points.

We thank Reviewer #1 for the opportunity to clarify this very important point. To clarify from the start, the study has not selected or removed any data point, there is no selection bias.

→ The reason why the number of replicates is different depending on the condition and/or genotype is due to reasons summarized here—and detailed in our reply to each Reviewer #1's comment—as follows: a) In each experiment, we tried to use littermates and, naturally, the number of *Zeb1*^{WT} and *Zeb1*^{ΔM} mice in a given litter are never the same; b) In each experiment and for each genotype, we tried to distribute mice (for *in vivo* experiments, which are most of the experiments in the study) or isolated macrophages (*ex vivo*) evenly across the different conditions. However, in many experiments, we allocated fewer mice for the "untreated" condition because *Zeb1*^{WT} and *Zeb1*^{ΔM} macrophages usually do not display phenotypic or functional differences under basal conditions ("untreated"); c) In any case, and with these limitations, in the revised version, we have tried to harmonize the number of replicates in each condition and genotype.

Fourth, in many cases single measurements are depicted, and neither quantification nor statistical analysis are performed

Once again, we appreciate the chance to clarify this issue and apologize for any possible confusion created.

It is important to note that none of the experiments in the original or revised manuscripts corresponds to single samples or measurements. Although some experiments may have appeared as single measurements, as noted in the figure legends, all experiments included multiple data points even if only the average value was shown.

→ Following the comments by this Reviewer, and as explained below in our reply to each specific comment by Reviewer #1, we have included single data points and have increased the number of replicates and samples in many of the figures, especially those that use human samples.

All of these issues make the paper appear immature. Further important problems following the sequence of figures.

Figure 1.

Importantly, to avoid any potential confusion, it should be noted that the data previously presented in Figure 2 is now presented in Figure 3.

A) The legend is insufficient. Exact numbers of mice have to be provided for each genotype and each condition separately. Furthermore, it needs to be stated in detail, how many mice died due to the intervention, and how often mice had to be euthanized. Any mouse that was not analyzed in full (i.e. was taken out of the analysis) has to be depicted. Please elaborate: What was your reason to use this number of mice, i.e. was there a predefined number. All of this information is essential!

We thank Reviewer #1 for her/his constructive suggestions and the chance to further clarify how the experiment was set up.

- The Committee on Animal Research and Ethics at our institution categorizes sterile toxic shock as a "very severe procedure". Consequently, mice must be monitored by two separate investigators for several parameters—including overall appearance, level of consciousness, locomotor activity, respiration rate, temperature, and body weight—every 2 h during the initial 12 h after the last LPS injection and every 4 h (except at night). At each monitoring time point, a wellness score is registered for each mouse and the ethical end-point is established when a mouse presents any of the following signs: absence of locomotion, difficulty breathing, body temperature below 30°C, or loss of body weight greater than 20%. This forced us to euthanize one (1) *Zeb1^{ΔM}* mouse in the LPS + LPS condition before the end of the protocol (see below).
- Based on similar experiments in the literature (*Nat Immunol* 19:561-570; *Nat Commun* 11:6343), we obtained the permission from our Committee on Animal Research and Ethics to use a maximum of 15 mice per genotype and condition. Eventually, 15 *Zeb1^{WT}* and 12 *Zeb1^{ΔM}* mice were subjected to LPS protocol (left panel of Figure 1A) and 14 *Zeb1^{WT}* and 12 *Zeb1^{ΔM}* mice were subjected to LPS + LPS protocol (right panel of Figure 1A).
- In both protocols (LPS, LPS + LPS), mice were followed up until the survival of one of the groups fell below 35%. In the LPS protocol (left panel of Figure 1A), out of the 15 *Zeb1^{WT}* mice included in experiment, 5 survived and out of 12 *Zeb1^{ΔM}* mice included in the experiment, 9 survived. In the LPS+LPS condition (right panel of Figure 1A), out of the 14 *Zeb1^{WT}* participating mice, 11 survived, while only 4 of the 12 *Zeb1^{ΔM}* mice survived.

- In the revised version, the number of mice by genotype at the start of the protocol and the number of mice that died or survived are specified. Please note that, due to the word limit of figure legends (maximum of 350 words), not all the information above can be included in the figure legend and most of the information is in the Supplementary Methods section.

B) The authors define peritoneal macrophages as CD45+ CD11b+ F4/80+. It would be desirable to see the gating strategies of all conditions: untreated, LPS, LPS+LPS. As there is probably beginning influx of inflammatory monocytes 24h post LPS treatment, we recommend to add Ly6C to the panel to rule out that there is monocyte contamination in their “peritoneal macrophages” if they compare them to the LPS 3h condition. Ideally, the authors should provide a panel that shows all myeloid cells that could express LysM (neutrophils, monocytes, dendritic cells), with respect to the later experiments.

We thank the Reviewer for his/her helpful suggestions.

The revised manuscript has addressed all the above comments and a) includes the FACS gating plots for the untreated, LPS and LPS + LPS conditions (Supplementary Figure S1M); b) analyzes Ly6C expression and includes a panel of myeloid populations under the three conditions (untreated, LPS and LPS + LPS) (Supplementary Figure S1L). Indeed, as noted by Reviewer #1, in the *in vivo* experiments, the treatment with a second dose of LPS led to an influx of Ly6C⁺ monocytes (new Supplementary Figure S1L). However, it should be noted that in all the *ex vivo* experiments (e.g., Figures 1C, 1D, 1E, 1G, 1I, 1J, 1M, 2F, 2G, 2I, 2J, 3A, 3B, 3C, 3D, 3H, 3I, 3J, 5A, Supplementary Figures S3A, S3B, S3C, S3I, S3J, S5A, S5D, S5F), we selected for CD11b⁺/F4/80⁺, thus excluding F4/80⁻ and F4/80^{low} monocytes (see FACS plot in Supplementary Figure S1M).

→ Following the suggestion of Reviewer #1, the revised version has implemented the following changes: a) it included FACS gating plots for the untreated, LPS and LPS + LPS conditions (Supplementary Figure S1M); b) it included a panel of myeloid populations either untreated, treated with LPS and treated with LPS + LPS (Supplementary Figure S1L).

B-D) It seems that Zeb1ko macrophages are non-responsive with regard to LPS stimulation (either one or two). It would be interesting to see transcripts, if present immunity related, that are actually upregulated under the Zeb1 knock out conditions. The depiction should be simplified by combining 1B and 1D. Instead, levels of multiple cytokines, not just IL6 should be measured e.g. by bead assays to complement the transcriptional analysis.

We thank the Reviewer for the constructive comments.

- We have now combined the heatmaps of former Figures 1B and 1D into a single figure (now Figure 1E).

- We would like to note that Figure 1E includes genes related to inflammatory responses that are downregulated in *Zeb1*^{ΔM} macrophages relative to *Zeb1*^{WT} macrophages in response to one or two doses of LPS. However, we have now included the pro-inflammatory gene *Il6st* in Figure 1E, indicating that not all inflammation-related genes were downregulated in *Zeb1*^{ΔM} macrophages. Additionally, as shown in the Suppl Figures S1N and S2A, most of the transcriptome outside this set of genes is not downregulated in *Zeb1*^{ΔM} macrophages with respect to *Zeb1*^{WT} counterparts.
- We also appreciate his/her suggestion to examine the protein levels of a panel of cytokines using bead assays. To that effect, we used a commercial multiplexed sandwich- and bead-based quantitative antibody array commercial kit (RayPlex® Mouse Inflammation Array Kit 1, RayBiotech Life, Inc.), which was assessed by FACS. We analyzed 8 mice per genotype and condition in pools of two mice for each analysis to reach cytokine levels above the sensitivity threshold of the kit. As shown in the new Figure 3H, the sensitivity of this array was able to detect 5 of the 13 cytokines included in the kit, namely IL1β, IL4, IL6, TNFα, and CXCL1. In the LPS condition, *Zeb1*^{ΔM} macrophages expressed lower levels of these cytokines than *Zeb1*^{WT} macrophages; in contrast, in the LPS + LPS condition, *Zeb1*^{ΔM} macrophages expressed higher levels of IL6 and CXCL1. No detectable levels of IL1β or TNFα were produced by macrophages of either phenotype in the LPS + LPS condition.

➔ In response to the suggestions by this Reviewer, the revised manuscript has implemented the following changes: a) the heatmaps of former Figures 1B and 1D have been combined into a single heatmap (Figure 1E); b) we have analyzed the protein levels of a panel of cytokines beyond IL6 using a bead-based quantitative antibody array (Figure 3H).

C + H) Alle western blots need to be analyzed by densitometry (including statistics of n=3 or more samples).

We thank Reviewer for her/his suggestion.

Western blots in the former Figures 1C and 1H (now Figures 1B and 2C), as well as of the Western blots of other figures in the paper, have been now quantified by densitometry. In addition, the number of samples in Figure 1B and 2C have been increased to a total “n” of 5.

➔ Following Reviewer #1's comment, the number of samples in Figures 1B and 2C has been increased and the quantification of the Western blots of five mouse samples and five human samples have been added to the right side of the representative blot.

E) The authors show a panel of transcripts that are involved in glycolysis. It would be nice if this panel could be completed, i.e. add remaining enzymes such as *Gapdh*, *Pgk1*, *Eno2*, *Pkm*, *Ldha*.

Following the comment by Reviewer #1, the heatmap of Figure 1H (former Figure 1E) now also includes the suggested genes (*Gapdh*, *Pgk1*, *Eno2*, *Pkm*, *Ldha*) as well as *Slc16a3*.

→ The heatmap of Figure 1H has been revised to include additional genes (*Gapdh*, *Pgk1*, *Eno2*, *Pkm*, *Ldha*, *Slc16a3*).

E) – F) The authors suggest that decreased levels of lactate play a role in the impaired anti-inflammatory transition of *Zeb1* knock out macrophages. They show the transcript of monocarboxylic acid transporter *Slc16a* member 1, however the member 3 (*Slc16a3*) might have a higher affinity for lactate and should be analyzed.

We appreciate the insightful comment raised by the Reviewer. As noted in response to the previous point, the expression of *Slc16a3* has been included to the heatmap of Figure 1H. In subsequent experiments (e.g., Figures 4A and 4B), lactate is included in experiment for its reported anti-inflammatory effects in macrophages treated with LPS (e.g., *Nature*, 574:575-80). Our results in Figures 4A and 4B suggest that the anti-inflammatory effect of lactate in LPS-treated *Zeb1*^{WT} macrophages is due to their *Zeb1*-mediated production of lactate rather than to its transport.

→ The expression of *Slc16a3* has been included in the heatmap of Figure 1H.

I) Here, and in all instances comparisons of all (!) data point of one experiment need to be statistically analyzed. The selection of comparisons seems random, this needs to be changed.

We appreciate the opportunity to clarify this important point.

All comparisons in the experiment were assessed for statistical significance. All Statistically significant comparisons were labeled with stars (* or **). In the original manuscript, non-significant comparisons were not labeled as “ns” to avoid overcrowding the figure. We thought that this was understood, but perhaps not including the “ns” has created more confusion. Therefore, in response to the Reviewer's request, all comparisons in the revised Figure 2F (former Figure 1I) and in most other figures throughout the manuscript include labels indicating their corresponding level of significance, including a 'ns' label for non-significant comparisons

→ The new Figure 2F includes labels indicating the corresponding level of significance of all comparisons, including a “ns” label for non-significant comparisons

J) Could the authors please provide a set of genes that are associated with autophagy/mitophagy or autophagy of mitochondria (similar to Figure S1K). This would relieve the impression of choosing only differentially expressed genes.

- We thank the Reviewer for her/his constructive suggestion. The revised Figure 2G (former Figure 1J) now includes a larger panel of autophagy/mitophagy-related genes (genes in the Gene Ontology Term GO:0000422).
- Regarding Supplementary Figure S1K; that figure corresponded to the analysis of a published array of wild-type bone marrow-derived macrophages (*Nature*, 574:575-580.), therefore is not possible to analyze the effect of *Zeb1* knockdown. See below for additional details.

→ Following the comment by this Reviewer, the heatmap of Figure 2G now includes additional autophagy/mitophagy-related genes in the Gene Ontology Term GO:0000422.

Figure S1

G) Please indicate conditions of the western plot

We thank the Reviewer for pointing out to this issue. The blots included in the original Supplementary Figure S1G (now Supplementary Figure S1O) are the unedited blots corresponding to Figure 1B. Since Supplementary Figure S1 is very crowded, we thought that the labeling of the blot was redundant as it is parallel to Figure 1B. But we concur with this Reviewer that this may have created some confusion. Accordingly, Supplementary Figure S1O (former Supplementary Figure S1G) is now labeled with all the conditions and genotypes.

→ Following the comment by Reviewer #1, the labeling of Supplementary Figure S2B (former Supplementary Figure S1O) specified all the conditions and genotypes.

J) Please indicate conditions of the western plot (right).

Once again, we thank the Reviewer for pointing out this issue. The blots on the right side of the original Supplementary Figure 1J (now Supplementary Figure S2B) are the unedited blots of blots on the left side. The revised figure now includes labeling for all the conditions.

→ Following the comment by Reviewer #1, the labeling of Supplementary Figure S2B (former Supplementary Figure S1J) specified all the conditions.

K) Please show color code of the heatmap.

We are grateful to Reviewer #1 for pointing out this oversight.

Regarding the former Supplementary Figure S1K, we would like to emphasize that this figure corresponded to the analysis of a published array of wild-type bone marrow-derived macrophages (*Nature*, 574:575-580. Therefore is not possible to analyze the effect of *Zeb1* knockdown. In the revised manuscript, we eliminated the former Supplementary Figure S1K or several reasons: a) both Reviewers indicated that the Supplementary Figure S1 was very

crowded; b) *Nature* 574:575-580 used bone marrow-derived macrophages instead of peritoneal macrophages, and c) as indicated by Reviewer #1, the heatmap is only a correlation.

K) To be able to fully grasp the differences observed in mitophagy under these conditions, the authors should provide a set of genes associated with autophagy of mitochondria (, e.g. GO term 0000422; which includes *Usp30* and *Sqstm1*). Also why are the chosen genes not analyzed in the *Zeb1* knock out condition? This might add a piece to the puzzle.

We thank the Reviewer for the chance to clarify the confusion created by former Supplementary Figure S1K.

As indicated in the main text and legend in the original manuscript, the heatmap in this Figure corresponded to “*the expression of selected genes in bone marrow- derived macrophages untreated or treated with LPS for different periods from the published array GSE115354*”. Therefore, since it is not our own RNAseq it is not possible to analyze the effect of *Zeb1* knockdown because the authors of the study (Zhang et al., *Nature*. 2019;574:575-580) conducted their RNAseq in wild-type bone marrow-derived macrophages. We eliminated this Figure in the revised manuscript because of several reasons: a) as both Reviewers indicated, Supplementary Figure S1 is very crowded; b) Zhang et al., *Nature*. 2019;574:575-580 used bone marrow-derived macrophages, and c) as indicated by Reviewer #1, the heatmap is only a correlation.

→ Following the suggestion by this Reviewer, the heatmap of Figure 1H now includes additional autophagy/mitophagy-related genes in the Gene Ontology Term GO:0000422.

→ Following Reviewer #1’s suggestion and in order to gain a deeper understanding of the variations observed in mitophagy during inflammatory conditions, we conducted two additional experiments

- We analyze the *SQSTM1* promoter and found the presence of consensus binding sequences for ZEB1 that we tested by chromatin immunoprecipitation (ChIP) assays. We found that ZEB1 bound to the *SQSTM1* promoter, and interestingly, binding was higher in human monocyte-derived macrophages treated with LPS compared to those treated with LPS+LPS (Supplementary Fig. S2E).
- We used a mitophagy detection commercial kit to detect lysosomes and their fusion with damaged mitochondria engulfed in autophagosomes in *Zeb1*^{WT} and *Zeb1*^{ΔM} macrophages under inflammatory and immunosuppression conditions (Supplementary Figure S2F).

L) Please show color code of the heatmap.

The revised figure (now Supplementary Figure S2A) includes the color code of the heatmap

L) Why does the macrophage induce mitochondria DNA-encoded gene transcription, i.e. increases the production of mitochondria content, to shortly after (~27h; Figure 1G) remove mitochondria from the cell via autophagy? Assuming that the decrease of mitochondria numbers would also happen without a second LPS administration. Could the authors please speculate.

Regarding former Supplementary Figure S1L (Supplementary Figure S2A in the revised manuscript). We appreciate the insights of this Reviewer and the suggestion to further investigate this issue. Along with the comments of Reviewer #2, we have now investigated the role of ZEB1 in the regulation of mitochondrial DNA content. We found that differences in mitochondrial DNA content between *Zeb1*^{WT} and *Zeb1*^{ΔM} macrophages are due, at least in part, to glutamine levels (new Figure 6H).

Collectively, our results would suggest that, during the inflammatory and tolerogenic responses, rapid changes in mitochondrial dynamics parallel those observed in cytokine production. We concur with Reviewer #1 that the decline in mitochondrial number occurs even without the second dose of LPS. The electron microscopy morphology data in Figure 2F (former Figure 1I) examines the effects of the second dose of LPS in comparison with a single LPS dose. In line with other studies (*Nat Immunol*, 12:222-30; *Nature*, 560:198-203), our data indicate that bacterial LPS triggers an increase in mtDNA replication and ROS production in macrophages. However, this response must be self-limited to avoid tissue and cell damage, thus triggering the activation of autophagy/mitophagy to remove damaged mitochondria, which are known to trigger an inflammatory response (*Cell*, 164:896-910). The revised manuscript elaborates on this self-regulatory process (page 11 in the Results section and page 26 in Discussion).

→ Following comments by both Reviewers, we have examined mitochondrial DNA content in *Zeb1*^{WT} and *Zeb1*^{ΔM} macrophages (new Figure 6H). The revised manuscript also discusses the increase in mtDNA replication and the induction of the autophagy/mitophagy pathway following LPS treatment (page 11 in the Results section and page 26 in Discussion).

M) Please indicate conditions of the western plot

The blots in the former Supplementary Figure S1M (new Supplementary Figure S2C) corresponded to the unedited blots of Figure 1H. We eliminated the labeling to simplify an already very crowded Supplementary Figure S1. In the revised manuscript, Supplementary Figure S2C is now labeled for all conditions and genotypes.

Figure 2.

Importantly, to avoid any potential confusion, it should be noted that the data previously presented in Figure 2 is now presented in Figure 3.

A) Is not referenced in the manuscript. Instead of Figure 2A, 2B is referenced.

We thank Reviewer #1 for alerting us about this issue, which has been now corrected

B) Should be referenced a sentence later.

This issue has been corrected in the revised version.

C) Analysis as above.

In the revised manuscript, the Western blots in the former Figure 2C have been quantified by densitometry. The Figure has been moved to the Supplementary section (new Supplementary Figure S3C) to simplify and reduce the size of Figure 2. Blots included in the left panel are representative of four independent experiments, which were quantified in the right panel

→ Following the suggestion by this Reviewer, the revised manuscript quantifies Western blots in the former Figure 2C (now the Supplementary Figure S3C).

D) E) What are the authors referring to as „OCR“ in the right graph? It would be helpful if the authors could also indicate the measured spare capacity in the figure or figure legend. How many cells did the authors use, as basal respiration seems quite low. The authors refer to ECAR in that section however, they do not show ECAR plots or quantification. I suggest to remove ECAR from the text.

We thank the Reviewer for the chance to clarify the labeling of former Figures 2D and 2E (now Figures 3B and 3C) and her/his suggestion to measure the respiratory spare capacity.

- In the original figures (former Figures 2B and 2C), the labeling “pmoles / min” referred to the units in which OCR was measured but the term “relative” was probably confusing. The term “relative” meant to indicate that the basal OCR in the untreated condition was arbitrarily set to 100 and the rest of conditions were expressed relative to the untreated condition. In the revised manuscript, Figures 3B and 3C, the graph bar is now labeled as the fold change relative to the “untreated” condition.
- Following the suggestion by this Reviewer, we have now measured the spare capacity in *Zeb1^{WT}* and *Zeb1^{ΔM}* macrophages in Figures 3B and 3C. In both genotypes, the spare capacity of the “untreated” condition to 100 and represent the rest of conditions relative to it.
- As noted in Supplementary Methods (page 20 in the original version, page 16 in the revised version), in the Seahorse experiments, 2×10^5 cells were seeded onto XF96 tissue culture plates. We agree with this Reviewer that basal respiration is low, but it is in line with that reported elsewhere in peritoneal macrophages (e.g., *Nat Commun*, 11: 3816).
- Regarding ECAR. We thank the Reviewer for pointing this out and the term “ECAR” has been removed from the text.

We thank the Reviewer for point out to the reference to ECAR in the original manuscript.

→ Following the comment by this Reviewer, the revised manuscript has implemented the following changes: a) the panels showing OCR levels (Figures 3B and 3C) have been relabeled; b) the spare respiratory capacity has been measured (Figures 3B and 3C); and c) the term ECAR was removed from the text.

F) The following controls are missing in vivo and need to be provided: The vehicle control for metformin and LPS, and metformin alone. These controls are very important (injections are a treatment themselves). These data are very important as a basis for the transcriptome analysis in Fig. 2i.

We are grateful for the opportunity to clarify this very important issue.

- Inclusion of the condition “Metformin alone”. As noted at the start of this letter, in the absence of LPS, metformin has no effect on any marker of inflammation. Reviewer #1 is correct that this information should have been shown in Supplementary Fig S1, but we did not include it in the original manuscript because that Figure was too crowded. In the revised version, we have included the condition of “metformin alone” in the first inflammation-related *in vitro* and *in vivo* experiments where metformin was used (Figures 3D, 3G, 3H and 5F).
- Consequently, we compared the transcriptome of *Zeb1*^{WT} and *Zeb1*^{ΔM} macrophages treated with a dose of LPS preceded by a dose of either PBS or metformin (Figures 3I and 3J). The goal of this analysis was to assess the changes in gene expression during inflammation produced by the pre-treatment with metformin.
- Inclusion of the condition “Vehicle control”. As also briefly discussed at the beginning of this letter, mice received two injections. Untreated mice and mice receiving only one compound (metformin alone or a single dose of LPS) were injected with PBS in lieu of the missing dose(s) of the compound (metformin or LPS). The same applies to the IMQ mouse model of psoriatic disease in Figure 5. The use of PBS as vehicle control was specified in the original Supplementary Methods. Due to the strict word limits (350 words) established by Nature Communications for figure legends, it was not possible to include all this information. Nevertheless, the revised version of the main text (page 7) and Supplementary Methods (pages 5-6) stresses the injection of vehicle as control.

Line 292, referring to Figure 1A, instead it should refer to Figure 1B

We thank the Reviewer for pointing this out. The mistake has been corrected in the revised version.

G) Again, essential controls are missing (see 2F), in this case even *Zeb1*ΔMac untreated and LPS only condition. Accordingly, „thus, eliminating the ZEB1-mediated inflammatory difference between both genotypes showed above in Figure 1B (Fig. 2G) is overstated given the fact that controls are missing and „only“ IL-6 transcript is measured compared to an inflammatory gene set in Figure 1B.

We appreciate the chance to strengthen this figure. IL6 expression is now shown under all experimental conditions. The new data are shown in the new Figures 3D, 3G and 3H, which replaces the previous Figure 2G.

→ Following Reviewer #1's comments, the mRNA levels and protein expression of IL6 in all conditions (untreated, metformin alone, LPS, metformin + LPS, and LPS + LPS) is now shown in included in Figures 3D, 3G and 3H, which replaces the original Figure 2G.

Figure S2.

A) Please indicate conditions of the western plot [sic]

Overall: Single measurements are not acceptable. Quantification (densitometry) and statistical analysis is absolutely required).

Thank you for allowing us to clarify this issue.

The blots in the former Supplementary Figure S2A (currently Supplementary Figure S3C) are the unedited blots corresponding to the former Figure 2C (now Supplementary Figure S3C). Supplementary Figure S3C has now been labeled for all conditions.

Neither the original manuscript nor the revised manuscript included single measurements or single western blots. Western blots shown in former Supplementary Figure S2A are representative of four independent experiments. In the revised manuscript, Supplementary Figure S3C now includes the quantification of the four experiments.

Figure 3.

Importantly, to avoid any potential confusion, it should be noted that the data previously presented in Figure 3 is now presented in Figure 4.

A) The GSEA are not in line with what is measured in Figure 3SE. While there is a clear difference in ROS production visible in the in vivo luminescence assay, there is no significant difference in the gene sets of Zeb1wt and Zeb1ΔMac treated with a single dose of LPS.

Vice versa, only one of the two representative mice of Zeb1wt in Figure 3E shows a luminescence signal. Suggesting less ROS production in Zeb1wt, what we would expect given the results from Figure 3 B), but opposing a significant difference in the GSEA of 2xLPS treatment in the „favor“ of Zeb1wt in this subfigure. Also, I would advise to only state the genotype underneath the enrichment score of the GSEA and better not add the pre-treatment condition as well. The treatment conditions in the top line are sufficient and „MET LPS in Zeb1ΔMac MET“ is confusing in my opinion.

We thank the Reviewer for her/his comments on the need to better explain former Figure 3A and Supplementary Figure S3E.

- We would like to clarify that the annotation “GOBP Superoxide metabolic process” in Figure 4A (former Figure 3A) is a metabolic signature including mostly anti-oxidant enzymes (e.g., *Cbs*, *Ccs*, *Cyb5r4*, *Nnros*, *Slc1a1*, *Sod1*, *Sod2*, *Sod3*, etc.). The GSEA plot for this annotation in LPS-treated *Zeb1^{WT}* versus *Zeb1^{ΔM}* macrophages was consistent with Figure 4B and Supplementary Figure S4E.

- Nevertheless, as the Reviewer correctly pointed out, the GSEA plot for *GOBP Superoxide metabolic process* in LPS-treated *Zeb1^{WT}* versus *Zeb1^{ΔM}* macrophages in former Figure 3A was not statistically significant. Consequently, it has been eliminated in the revised manuscript.
- We are also grateful to Reviewer #1 for her/his comments to strengthen Supplementary Figure S4E (former Supplementary Figure S3E). We have implemented the practice of shaving mice before bioluminescence assessment, which has resulted in improved signal detection. We have included new *in vivo* bioluminescence images for LPS and LPS + LPS. On the right side of Supplementary Figure S4E, we have also included the quantification of five mice per genotype and condition.
- Finally, and following the suggestion by this Reviewer, we have also modified the display of GSEA plots in Figure 4A (former Figure 3A) and other figures in the manuscript showing GSEA plots.

→ Following Reviewer #1's comments, the revised version has implemented the following changes: a) In Figure 4A, the GSEA for the condition "*GOBP Superoxide metabolic process*" in LPS-treated *Zeb1^{WT}* versus *Zeb1^{ΔM}* macrophages has been eliminated; b) new *in vivo* bioluminescence pictures and the quantification of five mice per genotype and condition have been included in Supplementary Figure S4E, and c) the display of GSEA graphs in Figure 4A and other figures with GSEA graphs throughout the manuscript has been modified to improve their clarity.

B) "Metformin reduced ROS production in *Zeb1^{WT}* macrophages but not in *Zeb1^{ΔM}* Mac ones" This statement is wrong: Metformin increases ROS production in *Zeb1^{ΔM}* Mac! Please explain.

We thank Reviewer #1 for her/his comment and the possibility to discuss this point.

Indeed, as noted by Reviewer #1, compared to macrophages from *Zeb1^{WT}* mice, the macrophages of *Zeb1^{ΔM}* mice treated with metformin displayed enhanced ROS production (Figure 4B) and amino acid transporter expression (Figure 6C).

It has been shown that treatment of macrophages with metformin reduces inflammation by inhibiting the mitochondrial ATP production and ATP-dependent mtDNA synthesis, which results in reduced oxidized mtDNA and inhibition of NLRP3 inflammasome activation (*Immunity*, 2021;54:1463-1477). As indicated in Figure 5H, metformin has no effect on mtDNA inhibition in *Zeb1^{ΔM}* macrophages. Likewise, in cancer cells, metformin upregulates ROS via AMPK-FOXO3a-SOD2 (*Sci Rep*, 2021;11:14002). We have previously reported that ZEB1 directly activates the transcription of FOXO3 in muscle stem cells (*Nat Commun*, 2019;10:1364). We hypothesize that in *Zeb1^{ΔM}* macrophages, metformin could activate the NLRP3 inflammasome via increased levels of oxidized mtDNA. Nonetheless, it cannot be ruled out that low levels of *Foxo3a* in *Zeb1^{ΔM}* macrophages are involved in the oxidative effect of metformin.

Furthermore, it is not acceptable that for each in vitro condition a different number of measurements are performed (or just depicted). In all experimental conditions the n has to be similar (at least 3 independent measurement).

We are very grateful for the opportunity to clarify this critical issue.

As noted at the start of this letter, we did not cancel any data point in the Figure 4B (former Figure 3B) or any other figure in the article.

Instead, differences in the number of mice in each experimental condition are due to the following reasons: a) In each experiment, we tried to use littermates and, naturally, the number of *Zeb1*^{WT} and *Zeb1*^{ΔM} mice in a given litter is never the same; b) although we tried to allocate a similar number of mice of each genotype across the different conditions, we intentionally use fewer mice in the “untreated” condition. The motive for this is that, as can be seen in most figures in the study, *Zeb1*^{WT} and *Zeb1*^{ΔM} macrophages or mice do not display phenotypic or functional differences under basal conditions (“untreated”); c) in any case, and with these limitations, in the revised version, we have tried to harmonize the number of replicates in each condition and genotype.

D) Line 357 refers to Figure 3E instead of D

We thank the Reviewer for pointing this out. The mistake has been corrected in the revised version.

E) One patient? Please measure at least 5-10 patients, otherwise this figure has to be deleted.

We appreciate the possibility to clarify this very important point.

As noted in the previous section, neither the original manuscript nor the revised manuscript included experiments using single samples. Figure 3E is a representative FACS plot of at least five individuals (healthy, septic patients).

To dispel any confusion, in the revised manuscript, we have included the quantification of the MitoTracker™ Green (MTG) and Tetramethylrhodamine-methyl ester (TMRM) median fluorescence intensity (MFI) of 5 healthy individuals and 5 septic patients (at both time 0 h and 72 h) (new Figure 4E).

→ In the revised manuscript, the number of individuals and septic patients has been clarified by including graph bars with quantification of the MFI of MTG and TMRM.

F) See E).

Thanks for the possibility to clarify the number of experiments in the former Figure 3F (now Figure 4F)

As indicated in the legend for this figure in the original manuscript, the blot shown was representative of three independent experiments. We have now conducted two additional experiments and, in addition to the blots (Figure 4F) we now include the quantification of the five experiments (Figure 4G).

→ We have included a quantification of the five experiments for which the blots of Figure 4G (former Figure 3F) are shown.

H) In the manuscript the authors refer to a „emetine alone“ condition, but there is no such condition in Figure 3H.

We appreciate the chance to explain the experiment. Even if it was not shown, the condition with “emetine alone” in the former Figure 3H (now Supplementary Figure S4I) is always included. We used emetine to inhibit cytosolic translation and assess exclusively mitochondrial translation over which total translation is measured. Attending to the comment of Reviewer #1, the FACS plot for “emetine” alone has been included. As this figure is only an internal control for Figure 4I, the figure has been moved to the Supplementary section (Supplementary Figure S4I)

Figure S3.

D) The authors note that a drop of ATP production is necessary to produce ROS. Accordingly, ATP production in *Zeb1^{wt}* mice is less than in *Zeb1^{ΔMac}* after single LPS treatment. However, compared to the untreated controls the *Zeb1^{wt}* macrophages remain unchanged and the *Zeb1^{ΔMac}* macrophages drop in ATP production.

We appreciate the insight of Reviewer #1 regarding this issue. A possible explanation for this observation is that, compared to *Zeb1^{WT}* counterparts, *Zeb1^{ΔM}* macrophages treated with LPS displayed enhanced consumption of glucose (new Figure 6D) and glutamine (new Figure 6E), which should increase ATP production, potentially compensating (at least partially) the dampening of ATP triggered by increased ROS production.

E) To visualize the overall low luminescence signal better, it might help to shave the fur of the mice. Overall: Single measurements are not acceptable. Quantification (densitometry) and statistical analysis is absolutely required

We apologize for any potential confusion created by the data shown in Supplementary Figure S3E. In any case, we would like to clarify that neither this figure nor any other figure in the article represents single measurements. In this case, the images were representative of five mice per genotype and condition. On the other hand, we appreciate the Reviewer's suggestion to remove the fur from the mice, as it has significantly enhanced the signal quality. The bioluminescence quantification of these mice is now shown in the right panel of the new Supplementary Figure S3D.

→ Following the comments of this Reviewer, the new Supplementary Figure S4E includes a panel with the quantification of the bioluminescence signal.

Figure 4.

Importantly, to avoid any potential confusion, it should be noted that the data previously presented in Figure 4 is now presented in Figure 5.

C) Right panel: How do peritoneal macrophages contribute to the lactate formation as a mediator of psoriasis in the skin? Or why did the authors choose to look at peritoneal macrophages? I would suggest to have a new subfigure and do not put that data into the ear histology measurement data.

- We thank Reviewer #1 for the chance to clarify the use of peritoneal macrophages in the IMQ model of psoriasis. Like other mouse models of psoriasis, IMQ-induced skin inflammation is accompanied by systemic inflammation. Accordingly, other studies (e.g., *Cell* 177:1201-16; *EMBO Mol Med*, 14:e14455) have used peritoneal and/or bone marrow-derived macrophages as a proxy because the low number of macrophages in the ear skin hinders the phenotypic and/or functional analysis of macrophages). In fact, *Cell* 177:1201-16 uses bone marrow-derived macrophages to assess lactate secretion in IMQ-induced psoriasis. Supporting these articles, the results shown in Supplementary Figures S5A to S5D suggest that peritoneal macrophages serve as a reliable proxy for systemic inflammation.
- We are also grateful to Reviewer #1 for her/his suggestion to separate the data on lactate levels from histology measurements. In the revised version we split the original Figure 4C into two Figures, Figure 5C (epidermis and ear thickness) and Figure 5D (lactate levels). Supplementary Figure S5D shows that lactate and doxycycline reduce systemic inflammation. Figure 5C and Supplementary Figure S5D show that peritoneal macrophages from these models increase lactate production and decrease ROS production.

→ Following the comments by this Reviewer, former Figure 4C has been split into the new Figures 5C and 5D.

G) Again, the “metformin only” condition is missing.

As discussed above in our reply to Reviewer #1's comments on the former Figure 2F, the condition “metformin” alone was included in our experiments; however, it was not included in the original figures because metformin by itself has no effects on inflammatory markers and to simplify the display of the already large figures in the manuscript. However, we concur with the Reviewer that it is a relevant control and, in the revised manuscript, we have included back the condition “metformin” alone in the first *in vivo* and *in vitro* experiments where metformin was used. Thus, the condition “metformin” has been included in Figure 5F.

→ Following this Reviewer suggestion, in the revised manuscript, the condition “metformin” alone is now showed in Figure 5F.

l) Again, only one patient, i.e. not acceptable.

We appreciate the possibility to clarify the number of individuals analyzed in that experiment.

As noted with respect to other figures, neither the original manuscript nor the revised manuscript includes figures that use single samples. The right panel of the former Figure 4I (now Figure 5J) is only a representative FACS plot of a healthy individual and a PsA patient. But, as the right panel of the figure shows, the median fluorescence intensity of MTG was assessed in six healthy individuals and six patients with psoriatic arthritis (PsA). In addition, the mean fluorescence intensity of 6-carboxy 2',7'-dichlorodihydrofluorescein diacetate (CH₂DCFDA) was assessed in four healthy individuals and four PsA patients (Figure 5K). In the revised manuscript, the legend has been rewritten to clarify this point (page 44 of the revised manuscript).

→ In the former Figure 4I (now Figure 5J), the FACS plot of MTG is a representative plot of six healthy individuals and six PsA patients. In turn, the FACS plot of CH₂DCFDA is representative of four healthy individuals and four PsA patients (Figure 5K). In the revised manuscript, the legend has been rewritten to clarify this point (page 44 of the revised manuscript).

Figure S4.

D) E) It would be nice to see a *Zeb1*^{ΔM} untreated spleen quantified and as a picture.

We thank the Reviewer for her/his suggestion. In the revised manuscript, a representative picture of the spleen of an untreated *Zeb1*^{ΔM} mouse is now shown in the new Supplementary Figure S5F. Furthermore, quantification of the spleen weight from four untreated *Zeb1*^{ΔM} mice is shown in the right panel of the new Supplementary Figure S5F.

→ Following the comments by Reviewer #1, the new Supplementary Figure S5F includes the picture of the spleen of an untreated *Zeb1*^{ΔM} mouse as well as the quantification of the spleen weight from four untreated *Zeb1*^{ΔM} mice.

Figure 5.

Importantly, to avoid any potential confusion, it should be noted that the data previously presented in Figure 5 is now presented in Figure 6.

B) Again, highly variable numbers of repetitions are shown for each condition in vitro. This is unacceptable.

We are very grateful the chance to clarify how these experiments were performed.

As noted with respect to other figures, we are not canceling any data points. The reasons why the number of replicates is different for each condition are as follows. First, in each experiment, we aim to use littermates, but the number of *Zeb1*^{WT} and *Zeb1*^{ΔM} mice are never the same. Second, within each genotype, we always allocate fewer mice for the “untreated” condition than

for the LPS condition. The rationale for this is that *Zeb1*^{WT} and *Zeb1*^{ΔM} macrophages tend to be smaller or no phenotypic or functional difference under basal (untreated) conditions than under LPS or other treatments.

Given that: a) new data has been added to Figure 5 in the revised manuscript and the figure has become very crowded, and b) the former Figure 5B and 5C were negative results (they do not explain the phenotype), we have moved both figures to the Supplementary section as Supplementary Figures S6B and S6C.

E) Why do the authors use bone marrow-derived macrophages in this experiment? All the other experiments were carried out in peritoneal macrophages and glucose and glutamine uptake could also be performed in peritoneal macrophages.

The reason for using bone marrow-derived macrophages in the experiment of glucose and glutamine uptake is that, for all types of metabolomic analyses (e.g., amino acid uptake, ¹³C fluxomics, etc.), the relatively low metabolic rate of peritoneal macrophages means that many metabolites are below the detection threshold.

- In any case, we have now assessed the glutamine/glutamate and BCAAs content in peritoneal macrophages using a luminescence detection kit, which is more sensitive (new Figures 6F and 6G).
- We have also measured by liquid chromatography–mass spectrometry the levels of glucose and glutamine uptake in peritoneal macrophages (new Figures 6D and 6E).

→ Following the comments of both Reviewers, the revised manuscript has analyzed glucose and glutamine uptake (new Figures 6D and 6E) as well as the intracellular levels of glutamine/glutamate and BCAAs in peritoneal macrophages (new Figures 6F and 6G).

C) REPLY TO REVIEWER #2

Review of the paper “Metformin depends on ZEB1 expression in macrophages for its anti-inflammatory effects” by Cortes et al.

The study by Cortes et al., is focused on the role of the transcription factor ZEB1 as a major regulator of both acute macrophage activation and their transition to an immune tolerant/inflammatory resolution state. Using both mouse and human cells, the authors propose a mechanism in which under conditions of acute inflammation, Zeb1 increases glucose metabolism to control inflammatory gene expression. In contrast, under tolerogenic states, such as re-exposure of macrophages to LPS or Psoriasis, the authors describe a scenario in which Zeb1 expression limits amino acid uptake via control of the expression of glucose and amino acid transporters, leading to the suppression of mTORC1 signaling, reduced translation of mitochondrial biogenesis, induction of autophagy, increased lactate production, and the expression of anti-inflammatory genes. Lastly, the authors found that metformin treatment mimicked macrophage tolerance, and Zeb1 was required for the anti-inflammatory effects of metformin.

Overall, we find that study of the role of Zeb1 as a major regulator of acute activation and tolerance in macrophage to be novel and of high importance to the field of innate-immunity and immunometabolism. However, as written and presented the manuscript is convoluted, hard to follow, and the data does not fully support the authors conclusions. A primary critique of the manuscript is the experiments are generally missing key details, some main findings are not statistically significant, relies on superficial analysis to draw main conclusions, and does not use multiple approaches to support or further test key findings. Additionally, we find the manuscript to be confusing to follow, since the writing is not very clear, lacking in many cases conclusions or explanations of the main findings. We find it may help significantly improve the manuscript if the authors increase the quality of the data versus the quantity of the data. For example, increasing the depth of the study by focusing on a few strongly supported mechanisms/pathways vs those that are weakly supported. Please see below for other major and minor concerns:

We appreciate the positive comments of Reviewer #2 on the novelty and importance manuscript. We fully concur with her/his comments that the original manuscript was convoluted. The main reason for that is that the manuscript was under review in other journal where the referees requested unconnected experiments in multiple directions. The revised manuscript has been streamlined and we focused on the deepening the mechanism by which ZEB1 regulates inflammation.

→ Importantly, to avoid any potential confusion, it should be noted that the data previously presented in former Figures 2, 3, 4, and 5 are now presented in Figures 3, 4, 5, and 6, respectively.

Major:

1) For in vivo studies do the authors see any sex differences in the phenotypes between males and females? Please include the age and sex of all mice used in the study.

We appreciate the opportunity to explain this point and apologize if this was not clear in the original manuscript.

Given that figure legends in *Nature Communications* have a word limit of 350 words, information about the sex of mice had to be relegated to the Supplementary Methods section. As noted in the original manuscript, the experimental mouse model of LPS-induced septic shock was conducted in 8-10 week-old female mice, which have been shown to have a higher number of peritoneal macrophages (*Blood*, 118:5918-5927). For *ex vivo* experiments, macrophages were isolated from 6-10 week-old female mice. For the IMQ-induced mouse model of psoriasis, 10-12 week-old male mice were used. When we used and compared both male and female mice in the same experiments (e.g., in vivo determination of ROS, Supplementary Figure S5E), we found no difference between both sexes.

→ In the revised manuscript we have added a note in the legend of Figures 1A (page 40 in the revised manuscript, the first figure using the in vivo LPS models) and Figure 5A (page 45 in the revised manuscript, the first figure using the in vivo IMQ model), stating that the same applies to the remaining figures in the study.

2) For *in vitro* studies (including supplemental data), please include cell type used (BMDMs, peritoneal macrophages, etc.) clear stimulation details such as dose and timing of stimuli.

Again, we appreciate the opportunity to explain this issue and apologize if this was not clear in the original manuscript. With the exception of the former Figure 5E, which was carried out with bone marrow-derived macrophages, all other *in vitro* experiments in the study used peritoneal macrophages.

→ In the revised manuscript, and following comments by Reviewer #1 to conduct all the experiments in peritoneal macrophages, the experiment in the former Figure 5E has been now conducted in peritoneal macrophages (Figures 6D-6F). All details about the timeline and dose of compounds used in these experiments are included in Supplementary Methods (pages 3 and 7-8).

3) Please properly label all Western blots including those in supplemental such as Figure S1G, S1J, etc.)

We thank the Reviewer for her/his suggestion. In the original manuscript, blots in Supplementary Figures that corresponded to the uncropped version of Western blots in the main section were not labeled to simplify the already cramped figures. But we concur with the Reviewer that this may have been confusing.

→ In the revised version, all Western blots in the main and Supplementary Figures have been labeled.

4) In figure 1, both *in vitro* and *in vivo* experimental conditions (acute and immunosuppression conditions), can the authors carefully measure *Zeb1* expression kinetics for both transcript and protein levels? This was done in S1K, only under acute LPS exposure. In S1K we see that acute LPS reduces *Zeb1* expression by 24 hours, thus if *Zeb1* expression is down-regulated how is it also involved or necessary for LPS tolerance upon re-exposure to LPS?

We believe there has been some misunderstanding. The former Supplementary Figure S1K is not our own RNAseq. As noted in the legend of that Figure, it is our analysis of the published array GSE115354, which was conducted in wild-type bone marrow-derived macrophages, not in peritoneal macrophages (*Nature*, 574:575-580). Naturally, the authors of the paper did not examine the effect of *Zeb1* knockdown. The RNAseq has been deleted in the revised manuscript.

Furthermore, we thank the Reviewer for her/his suggestion to examine the ZEB1 mRNA and protein expression in mouse peritoneal macrophages. In the revised manuscript, we showed that in the progression from acute inflammation to immunosuppression, ZEB1 mRNA and protein levels increase not only in mouse peritoneal macrophages (new Supplementary Figure S1G-H) but also in human peripheral blood-derived macrophages (Supplementary Figure S1F and S3D).

5) For Fig1 and FigS1, cytokines such as IL-6 should be validated by ELISA.

We also appreciate Reviewer #2's suggestion to quantify IL6 by alternative methods. Following the suggestion by this Reviewer and Reviewer #1, we have examined the protein levels of a panel of cytokines using a commercial multiplexed sandwich- and bead-based quantitative antibody array commercial kit (RayPlex® Mouse Inflammation Array Kit 1, RayBiotech Life, Inc.), which can be assessed by FACS.

As shown in the new Figure 3H, the array was able to detect 5 of the 13 cytokines included in the kit, namely IL1 β , IL4, IL6, TNF α , and CXCL1. In the LPS condition, *Zeb1* ^{Δ M} macrophages expressed lower levels of the five cytokines than *Zeb1*^{WT} macrophages; in contrast, in the LPS + LPS condition, *Zeb1* ^{Δ M} macrophages expressed higher levels of IL6 and CXCL1, but no detectable levels of IL1 β or TNF α were produced by macrophages of either phenotype.

→ In response to this Reviewer's and Reviewer #1's suggestion, we have analyzed the protein levels of a panel of cytokines using a bead-based quantitative antibody array (Figure 3H).

6) Lines 265-267: “We hypothesized that metformin-mediated inhibition of ETC-I could mimic the immune tolerance in macrophages after two doses of LPS” Why?

We thank Reviewer #1 for the chance to elaborate on the hypothesis. Our data in Figure 11 suggest that the decline in mitochondria after two doses of LPS may be responsible for the deficient immune response (immunosuppression) in that condition. This led us to investigate the effect of other drugs known to inhibit mitochondrial function. Since, at the doses used in our study, metformin inhibits ETC-1, we hypothesized that its reported anti-inflammatory effects may mimic the immunosuppression effect of two doses of LPS. In the revised manuscript, the hypothesis has been rewritten to better convey its intended meaning (page 11-12).

→ Following Reviewer #2's comments, the hypothesis has been rewritten (page 11-12).

7) Figure 2DE-Not consistent with literature. Mouse macrophage shut down OCR after LPS treatment. What are the experimental conditions and why does this data contrast with the well-established finding in the field that LPS inhibits the electron transport chain and OCR?

We agree with the comments raised by Reviewer #1. In fact, we were also initially surprised to see that, under our experimental conditions, the administration of LPS did not reduce OCR in macrophages as described in some other papers.

However, upon a more exhaustive review of the literature, we found that other articles have also described that LPS increases OCR in mouse peritoneal and bone marrow-derived macrophages as well as human macrophages. For instance, LPS increases OCR in mouse peritoneal macrophages (Figure 4C of *Nat Commun* 11:3816; Figure 1A of *Cell Metab* 29:1003-1011; Figure 9A of *Nutrients* 8:215), mouse bone marrow-derived macrophages (Figure 4D of *Cell Mol Immunol* 19:504-515; Figure 5E at 3h of *Nat Commun* 13:6320) and human monocyte-derived macrophages (Supplementary Figure 2A of *Nat Commun* 8:16040).

Looking at the experimental conditions in all of these papers, we were unable to find the reason why some studies found LPS to reduce OCR, while others (including our own study) found LPS to increase OCR. Nevertheless, the authors in *Nat Commun* 13:6320 found that at 3 h stimulus of LPS upregulates OCR in mouse bone marrow-derived macrophages while at 16 h stimulus reduces it.

8) Line 281: Typo, should say “To test whether the *in vivo* anti-inflammatory effects of metformin depended on Zeb1...”

We thank the Reviewer for pointing out to this oversight. The mistake has been now corrected in the revised manuscript.

9) Line 306 refers to Figure 2J, which is not included in the manuscript data figures.

Once again, we are grateful to Reviewer #2 for pointing to this oversight. The sentence was intended to refer to the former “Figure 2H” (IL6 expression). The mistake has been now corrected in the revised manuscript.

10) Line 325-329. Confusing paragraph. What is the main point? Please clarify.

We apologize if the paragraph was not clear. The beginning of that section has been rewritten (page 14 in the revised manuscript).

The paragraph meant to explain that in the late stages of an acute systemic inflammatory, oxidative stress triggers apoptosis in immune cells, which contributes to the subsequent immunosuppression stage (reviewed in *Cell Death Dis.* 10:782). On the other hand, in the context of acute systemic inflammatory, autophagy serve as a counter- and self-limiting mechanism to overcome the effects of apoptosis.

11) Line 330-332: As stated above, please be clear in the paper that these are peritoneal macrophages.

We thank Reviewer #2 for the chance to clarify this point.

As noted in point 2) above, with the exception of the former Figure 5E that was carried out in bone marrow-derived macrophages, all other *ex vivo/in vitro* experiments the study were conducted with peritoneal macrophages.

In the revised manuscript, and following comments by Reviewer #1 to conduct all the experiments in peritoneal macrophages, the experiment in the former Figure 5E has been now also conducted in peritoneal macrophages (new Figures 6D-F).

12) Line 345-351: Instead of saying we validated, we suggest the authors explain how they tested their hypothesis. Otherwise, the experiment sounds biased or predetermined.

We appreciate Reviewer #2's suggestion. In the revised manuscript, the sentence has been rewritten (page 15, last paragraph).

13) 361-366: This paragraph is a summary of the observations. However, the authors should provide more insight and the implications of these observations to help the reader better understand what this means in regards to the role of Zeb1 in these processes.

We are grateful Reviewer #2's suggestion. In the revised manuscript, the paragraph has been rewritten (page 16, second paragraph).

14) Line 371: S3H, MTCO1 is not the same image/exposure cropped in the main figure.

We thank Reviewer #2 for pointing this oversight. In the revised manuscript, the exposure of MTCO1 is now the same in Figure 4F and Supplementary S4H.

15) Line 380/Figure 3H: There is no emetine alone treatment.

We thank Reviewer #2 for pointing this out. Although the condition "emetine alone" was not included in the former Figure 3H (now Supplementary Figure S4I), it is always included to inhibit cytosolic translation and assess exclusively mitochondrial translation over which total translation is measured.

In the revised manuscript, the FACS plot for "emetine" alone has been included. As this figure is only an internal control for Figure 4I, the plots have been relegated to the Supplementary section (Supplementary Figure S4I).

16) Figure 3I/S3I: The difference in mitochondrial protein translation is modest and does not appear to be significant between WT and Zeb1 KO macrophages. Thus, there is no data supporting the conclusion of lines 385-387, which is a major finding the author claim Zeb1 regulates.

We appreciate the opportunity to clarify the inhibition of mitochondrial translation by ZEB1.

- We respectfully disagree that the differences in mitochondrial translation are modest and not significant. We have now included the quantification of mitochondrial translation in seven independent experiments with 2-3 mice per genotype (right panel of Figure 4I). Compared to the untreated condition, LPS and LPS+LPS reduced mitochondrial translation in *Zeb1*^{WT} by 18% and 23%, respectively, differences that are statistically significant. Differences in mitochondrial translation between LPS-treated *Zeb1*^{WT} and *Zeb1*^{ΔM} macrophages are also statistically significant. On the other hand, in *Zeb1*^{ΔM} macrophages, LPS leads to a 6% increase in mitochondrial translation, while the combination of LPS+LPS does not induce any significant changes.

- The above experiments indicate that mitochondrial translation decreases in *Zeb1*^{WT} during their progression from untreated to acute inflammation and immunosuppression. In contrast, the mitochondrial translation did not significantly change in response to LPS and LPS+LPS in *Zeb1*^{ΔM} macrophages. Altogether, the results suggest a role of ZEB1 in suppressing mitochondrial translation during inflammation and immunosuppression.
- We would like to emphasize that these changes are in mitochondria translation, not in total translation. These changes are also in line with or above those reported in other studies. For example, as shown in Figure 4C and Supplementary Figures S4H-I from *Cell*, 167(3):816-828, the article from which we obtained the protocol for assessing protein synthesis by incorporation of L-homopropargylglycine (HPG), changes in total translation (not mitochondrial translation) in proliferating established cell lines and mouse embryo fibroblasts are in the same range as those we observed for mitochondrial translation in macrophages.

→ Attending to the comments of Reviewer #2, the revised manuscript has introduced the following changes: a) to better illustrate differences in mitochondrial translation between *Zeb1*^{WT} and *Zeb1*^{ΔM} macrophages, we have modified the display of the FACS plots; b) included a graph bar quantifying the mean fluorescence HPG staining in seven mice per genotype and condition; c) to better convey the message of Figure 4I, the text has been rewritten (page 17).

17) Figure 3J/S3J: doesn't appear that ROS is statistically significantly between doxy-LPS WT and Zeb1 KO macrophage. Also, representative flow plot used in Fig 3J, does not match the MFI data in Fig S3J.

We thank Reviewer #2 for the possibility to improve Figure 4J and Supplementary Fig. S4J (former Figure 3J and Supplementary Figure S3J).

- Indeed, as the Reviewer indicates there is no statistical difference in ROS production between *Zeb1*^{WT} and *Zeb1*^{ΔM} macrophages in the Doxycycline + LPS. However, the difference between *Zeb1*^{WT} and *Zeb1*^{ΔM} macrophages in the LPS is statistically significant.
- We concur that the FACS plot in Figure 4J may appear to not be representative of the results in Supplementary Figure S4J. We have replaced the FACS plot by another one that is around the average of the data in Supplementary Figure S4J.

→ Following the comments by this Reviewer, we have replaced the FACS plot in Figure 4J.

18) Line 415-419: What's is the conclusion of this data? Please provide insight to what these observations mean.

We thank Reviewer #2 for the chance to clarify the sentence.

The sentence has been rewritten (page 18). The sentence meant to discuss our results in former Figure 4C (now Figure 5D) and Supplementary Figure S4A (now Supplementary Figure S5A) in the context of the reported anti-inflammatory effects of doxycycline and lactate (*Nature*, 574:575-580; *Immunity*, 54:53-67). Doxycycline reduces inflammation by dampening

mitochondrial translation, lactate functions by triggering anti-inflammatory and reparative responses.”

19) Figure 4 and SF4: IF staining is weak. It is hard to see colocalization of CD68 and ZEB1.

Following the comments by this Reviewer, the revised manuscript includes new captures of Figure 5I and Supplementary Fig. S5I (former Figure 4I and Supplementary Figure S4H) that show more clearly CD68 and ZEB1 staining.

→ Following Reviewer #2's comments, the captures of Figure 5I and Supplementary Fig. S5I have been replaced.

20) Line 440-446: Confusing paragraph, please make clearer.

We apologize that the message in the paragraph was not sufficiently clear. In the revised manuscript, the paragraph has been rewritten (page 19).

21) Lines 461-476: It was hard to follow why “ZEB1 inhibits mitochondrial protein translation during inflammation, through mechanisms independent of for CDK5RAP1 and mt-tRNAs modifications”. Please make this paragraph clearer and provide more insight and logic how this conclusion was made.

We apologize if the sentence was not sufficiently clear. The revised manuscript has rewritten the sentence (page 20).

The paragraph referred by Reviewer #2 attempted to explain that *Cdk5rap1* catalyzes ms2 modifications of mt-tRNAs for Ser(UCN), Phe, Tyr, and Trp, which are required for optimal mitochondrial translation (*Cell Metab* 21:428-42). The former Figure 5B showed that LPS downregulated *Cdk5rap1* expression in *Zeb1*^{WT} macrophages but not in *Zeb1*^{ΔM} macrophages. In contrast, ms2 modifications of the tRNAs of Ser(UCN), Phe, Tyr, and Trp were reduced in *Zeb1*^{ΔM} macrophages after one dose of LPS, but increase after two doses of LPS. Since ZEB1 regulates in opposing directions *Cdk5rap1* expression and ms2 tRNA modifications, they cannot explain the phenotype observed. Consequently, we have move the former Figures 5A, 5B and 5C to the Supplementary section as Supplementary Figures S6A, S6B and S6C.

→ In the revised manuscript, the sentence explaining the results on the regulation of *Cdk5rap1* and ms2 tRNA modifications by ZEB1 has been rewritten (page 20) and former Figure 5B and 5C have been moved to the Supplementary section as Supplementary Figures S6A, S6B and S6C.

22) Figure 5E: The data only shows that ZEB1 KO macrophages take up less glucose and more glutamine, but the data does not prove they consume or metabolize it.

We greatly appreciate the insights from Reviewer #2 and suggestions to strengthen the figure.

We concur with this Reviewer that data in the former Figure 5E (now Figure 6D) only showed the uptake of glutamine and glucose, but not necessarily their subsequent consumption/metabolization.

In addition to assessing glutamine and glucose uptake (former Figure 5E, now Figure 6D), in the revised manuscript we have conducted the additional experiments:

- We assessed glucose and glutamine levels in the supernatant of *Zeb1*^{WT} and *Zeb1*^{ΔM} macrophages treated with a single dose of LPS preceded by a dose of PBS, Metformin or LPS (new Figures 6D and 6E). A single dose of LPS increased glucose and glutamine consumption in *Zeb1*^{WT} but not in *Zeb1*^{ΔM} macrophages. In contrast, glucose and glutamine consumption were similar in *Zeb1*^{WT} and *Zeb1*^{ΔM} macrophages treated with a prior dose of metformin or LPS.
- We assessed the intracellular levels of BCAAs and glutamine and its metabolite glutamate in peritoneal macrophages (new Figures 6F and 6G). As for glucose and glutamine, a single dose of LPS increases the consumption of the BCAAs in *Zeb1*^{WT} macrophages, but not in *Zeb1*^{ΔM} macrophages. In contrast, intracellular BCAA levels were similar in *Zeb1*^{WT} and *Zeb1*^{ΔM} macrophages treated with a prior dose of metformin or LPS.
- The intracellular levels of glutamate were reduced in LPS-treated *Zeb1*^{ΔM} macrophages than in their *Zeb1*^{WT} counterparts. Of note, there was no difference in intracellular glutamine levels between LPS-treated *Zeb1*^{ΔM} and *Zeb1*^{WT} macrophages suggesting that, at least at the time analyzed, glutamine has been already metabolized to glutamate (Figure 6F).

→ Following the comments by Reviewer #2, we have assessed: a) glucose and glutamine levels in the supernatant of LPS-treated *Zeb1*^{WT} and *Zeb1*^{ΔM} macrophages under all experimental conditions (new Figures 6D and 6E); and b) intracellular levels of BCAAs, and glutamate and glutamine in *Zeb1*^{WT} and *Zeb1*^{ΔM} macrophages treated with one or two doses of LPS (new Figures 6F and 6G)

23) Line 481: Reference 50 does not show that SLC7A8/LAT2 “is required for the efflux of glutamine to incorporate leucine, thus activating mTORC1 signaling”. In fact, they argue the opposite.

We greatly appreciate the chance to clarify the point.

Indeed, as noted by Reviewer #2, the study in *Cell* 136:521-534 shows that LAT1 and LAT2 are required for the transport of glutamine and that incorporation of leucine activates mTORC1 signaling. Of note, several studies show that glutamine activates mTORC1 via SLC7A8 (*Nat Commun* 13, 6308; *J Exp Clin Cancer Res* 37:274).

We found that metformin or the first dose of LPS reduced the uptake of both glutamine and branched-chain amino acids (leucine, isoleucine, valine) in LPS-treated *Zeb1*^{WT} macrophages leading to mTORC1 inhibition (Figures 6E and 6F). Our results support that SLC7A8 uptakes branched chain amino acids (leucine, isoleucine, valine) and glutamine, which increases its intracellular levels, activating mTORC1. Nevertheless, we cannot rule out that other amino acid transporters are involved.

24) Figure 5F, very modest phenotype. Can the authors confirm SLC7A8 via western blot?

As shown in the former Figure 5F, the FACS plot for SLC7A8 expression corresponded to five independent experiments.

In the revised manuscript, we have assessed SLC7A8 protein expression in *Zeb1*^{WT} and *Zeb1*^{ΔM} macrophages under all experimental conditions by Western blot (new Figure 6C). In addition, we have also included a bar graph quantifying five independent Western blots (right side of Figure 6C). In line with the FACS analysis, the prior treatment with metformin or a first dose of LPS before the second dose of LPS reduced SLC7A8 in *Zeb1*^{WT} macrophages.

We concur with the Reviewer that changes in the FACS analysis of SCL7A8 are not only modest (as indicated in this comment #21) but also do not match with mRNA data (comment #22 above). The Western blot shown in the new Figure 6C has been conducted with commercial antibodies (Origene clone OTI 5A9, catalog number TA500503S for human samples and ImmunoGlobe catalog number 0142-10 for mouse samples), that are different from the antibody used in the FACS analysis in the former Figure 5F (Biolegend NAP-07, catalog number 368003). Since the new Western blot data concurs with both the RNA (qRT-PCR) and RNAseq data, we have decided to eliminate the FACS data.

25) Figure 5H. The qPCR data does not match the expression data presented in the flow data in Figure F. For example, LPS treated WT macrophages have significantly more SLC7A8 mRNA transcript compared to LPS treated ZEB1 KO macrophage, but less protein expression. Which means the metformin-LPS treatment qPCR data is not reliable as a measurement of SLC7A8 protein expression.

We fully concur with these comments by Reviewer #2. As she/he noted, the FACS data of the former Figure 5F does not match with the qRT-PCR quantification of RNA levels (former Figure 5H, now Figure 6B) and transcriptomics RNAseq data (former Figure 5D, now Figure 6A and Supplementary Figure S6D).

As noted in our reply to comment #21, the antibodies used in the FACS analysis and Western blots are different. Since the Western blot data in the new Figure 6C, does match with the qRT-PCR quantification and RNAseq data, we eliminated the FACS data.

26) Figure 5G: Besides mitotracker green, mitochondria content could be measured by looking at the expression of electron transport complexes and/or mito DNA copy number.

We thank Reviewer #2 for her/his insightful suggestions to strengthen this figure.

- Following the suggestion by this Reviewer, we have now measured mitochondrial DNA (mtDNA) copy number (MDCN) in *Zeb1*^{WT} and *Zeb1*^{ΔM} macrophages under the different experimental conditions and in the presence or absence of glutamine (new Figure 6H).
- In the presence of glutamine and compared to a single dose of LPS, pretreatment with metformin or two doses of LPS reduced MDCN in *Zeb1*^{WT} macrophages but not in *Zeb1*^{ΔM} macrophages.
- In contrast, in the absence of glutamine, LPS did not significantly alter MDCN. In the absence of glutamine, and compared to a single dose of LPS, metformin pre-treatment with metformin or two doses of glutamine did not change MDCN in either *Zeb1*^{WT} or *Zeb1*^{ΔM} macrophages.
- Overall, the results indicate that, when glutamine is available, the pre-treatment with metformin or a first dose of LPS reduces intracellular levels of glutamine and MDCN in LPS-treated *Zeb1*^{WT} but not in *Zeb1*^{ΔM} macrophages. As these changes do not occur in the absence of glutamine, our hypothesis is twofold: a) MDCN depends on the availability of glutamine, and b) glutamine levels in immunosuppressed macrophages are dependent on *Zeb1* expression.

→ Following the suggestion by this Reviewer, we have now assessed the mitochondrial DNA copy number in *Zeb1*^{WT} and *Zeb1*^{ΔM} macrophages under all experimental conditions and in the presence or absence of glutamine (new Figure 6H).

Minor:

1) Line 167: Insert Figure number 1D at the end of sentence. Also, it's not fully accurate to call these anti-inflammatory genes, MMPs for example can also be pro-inflammatory.

We are thankful for the suggestions. We would like to note that the former Figures 1B and 1D have been now combined into a single figure (new Figure 1E) and deleted the term anti-inflammatory.

2) Callout for Figure 2B should be Figure 2A (which is missing), and proper callout for Figure 2B should be at the end of the sentence in line 270.

We thank Reviewer #2 for pointing to this oversight. Please note that former Figure 2A is now Figure 3A. In the revised manuscript, the mistake has been corrected (page 12).

3) Figure 4D: straighten cropped images.

We thank Reviewer #2 for the suggestion. In the revised manuscript, the cropped images of ZEB1, MT-CO1, TOMM20 and GAPDH have been straightened (Figure 5E).

4) Line 479: typo

We thank Reviewer #2 for pointing out this typo. In the revised manuscript, the typo error has been corrected (page 21).

5) Line 481: typo

We are grateful to Reviewer #2 for pointing out this typo.

REVIEWERS' COMMENTS

Reviewer #1 (Remarks to the Author):

The authors have worked hard to address our concerns and suggestions. In particular, it's now much easier to follow the stream of arguments.

Remaining issues: The authors are still in part inconsistent in the statistical comparison of groups. Most notably in fig 1K: Conditions are compared across genotypes, yet not Zeb1dM-LPS vs. Zeb1wt LPS-LPS. It appears that Zeb1dM-LPS vs. Zeb1wt LPS-LPS is not different, and Zeb1dM-LPS vs. Zeb1dM-LPS-LPS is not different, I question that it is statistically sound that Zeb1wt LPS-LPS vs. Zeb1dM-LPS-LPS is different. This needs to be reviewed by a statistical expert or analysis has to be altered (to trend)

Fig S1M: the labels are not legible.

Reviewer #2 (Remarks to the Author):

The authors have done a great job of responding to the major critiques and concerns of the reviewers. The manuscript is significantly improved and suitable for publication.

September 22, 2023

RESPONSE TO REFEREES

Below is a summary of general changes introduced in the manuscript plus all comments made by the reviewers (in ***italic and bold***) followed by our response.

Below are the comments made by Reviewer #1 (in ***italic and bold***) followed by our response.

REPLY TO REVIEWER #1

The authors have worked hard to address our concerns and suggestions. In particular, it's now much easier to follow the stream of arguments.

We appreciate the positive comments of Reviewer #1 regarding the revised manuscript.

1) Remaining issues: The authors are still in part inconsistent in the statistical comparison of groups. Most notably in fig 1K: Conditions are compared across genotypes, yet not Zeb1dM-LPS vs. Zeb1wt LPS-LPS. It appears that Zeb1dM-LPS vs. Zeb1wt LPS-LPS is not different, and Zeb1dM-LPS vs. Zeb1dM-LPS-LPS is not different, I question that it is statistically sound that Zeb1wt LPS-LPS vs. Zeb1dM-LPS-LPS is different. This needs to be reviewed by a statistical expert or analysis has to be altered (to trend)

We are grateful to Reviewer #1 for her/his comments. Accordingly, the revised manuscript now includes the following changes.

- In response to Reviewer #1's comments, we consulted with Dr. C. Stephan-Otto Attolini, Head of the Biostatistics and Bioinformatics Service at the Barcelona Institute for Research in Biomedicine (IRB, Barcelona), and Dr. J. Rios, Head of the Biostatistics and Bioinformatics core facility at the August Pi Sunyer Biomedical Research Institute (IDIBAPS, Barcelona), who confirmed the suitability and adequacy of the statistical tests used in the study.
- While we conducted statistical analyses for all relevant comparisons, most figures indicate the statistical significance or lack thereof in comparisons between *Zeb1^{WT}* and *Zeb1^{ΔM}* samples (mice or macrophages) for a given treatment/condition, or between *Zeb1^{WT}* (or *Zeb1^{ΔM}*) samples across different treatments/conditions. Regarding Figure 1K and other figures, we did not include the statistical analysis for some comparisons where it would be challenging to ascertain whether the observed differences between groups were attributable to their distinct genotypes or different treatments. Nevertheless,

following this Reviewer's suggestion, and to ensure clarity and completeness without overcrowding the figures with brackets indicating statistical significance for all possible comparisons, we have included all statistical comparisons in the "Source Data" file. This file contains all the raw data along with the "p values" of statistical analyses for all figures in the study, not only Figure 1K.

- Following Reviewer #1's suggestion and Dr. Stephan-Otto Attolini's advice, the revised manuscript has expanded the labeling of statistical analyses in figures. In addition to using the notations *** ($p \leq 0.001$), ** ($p \leq 0.01$), and * ($p \leq 0.05$) for statistical significance, or denoting non-significance for p values > 0.05 , we have added numerical values above the brackets to indicate $0.05 < p < 0.075$. Furthermore, the "Source Data" file now includes the p -values for statistical analyses in all figures.

→ Following Reviewer #1's comments, the revised manuscript has incorporated the following changes: a) validation of the statistical tests in the study by an expert statistician, b) inclusion of "p values" in the "Source Data" file for all figures, and c) addition of numerical p values above brackets for comparisons with p values ranging from 0.05 to 0.075.

2) Fig S1M: the labels are not legible.

We are grateful to Reviewer #1 for pointing out this issue. We have added new labeling with higher font size in Supplementary Figure S1M.

REPLY TO REVIEWER #2

The authors have done a great job of responding to the major critiques and concerns of the reviewers. The manuscript is significantly improved and suitable for publication.

We appreciate the positive comments from Reviewer #2.